# Dietary restriction of cysteine and methionine sensitizes gliomas to ferroptosis and induces alterations in energetic metabolism

Pavan S. Upadhyayula [1,8], Dominique M. Higgins[1,8], Angeliki Mela [2,8], Matei Banu[1], Athanassios Dovas[2], Fereshteh Zandkarimi[3], Purvi Patel[4], Aayushi Mahajan[1], Nelson Humala[1], Trang T. T. Nguyen [2], Kunal R. Chaudhary[5], Lillian Liao[1], Michael Argenziano[1], Tejaswi Sudhakar[1], Colin P. Sperring[1], Benjamin L. Shapiro [1], Eman R. Ahmed[6], Connor Kinslow[3], Ling F. Ye[7], Markus D. Siegelin [2], Simon Cheng[3], Rajesh Soni[4], Jeffrey N. Bruce [1], Brent R. Stockwell [3,6] & Peter Canoll [2] ✉

Ferroptosis is mediated by lipid peroxidation of phospholipids containing polyunsaturated fatty acyl moieties. Glutathione, the key cellular antioxidant capable of inhibiting lipid peroxidation via the activity of the enzyme glutathione peroxidase 4 (GPX-4), is generated directly from the sulfur-containing amino acid cysteine, and indirectly from methionine via the transsulfuration pathway. Herein we show that cysteine and methionine deprivation (CMD) can synergize with the GPX4 inhibitor RSL3 to increase ferroptotic cell death and lipid peroxidation in both murine and human glioma cell lines and in ex vivo organotypic slice cultures. We also show that a cysteine-depleted, methionine-restricted diet can improve therapeutic response to RSL3 and prolong survival in a syngeneic orthotopic murine glioma model. Finally, this CMD diet leads to profound in vivo metabolomic, proteomic and lipidomic alterations, highlighting the potential for improving the efficacy of ferroptotic therapies in glioma treatment with a non-invasive dietary modification.

Glioblastoma is the most common malignant primary brain tumor with a median survival of only 16 months[1,2]. Glioma treatment resistance has been linked to oxidative stress and glutathione metabolism[3]. Oxidative stress, broadly defined as the (im)balance between reactive oxygen species and antioxidant defenses, underlies various distinct forms of cell death[4,5].

Ferroptosis is a form of regulated cell death that is iron dependent and mediated by lipid peroxidation. Glutathione, a reducing tripeptide with a thiol-containing cysteine residue,

serves as a cofactor for the enzyme glutathione peroxidase 4 (GPX4) to donate electrons to peroxides of polyunsaturated fatty acyl phospholipids[6–8]. Importantly, glutathione biosynthesis is dependent upon intracellular cysteine imported via the glutamate-cystine antiporter (system Xc-) and the enzymatic conversion of cysteine to glutathione. Methionine can also be converted to cysteine via the transsulfuration pathway to replenish glutathione. Therefore, ferroptosis inducers include compounds that inhibit system Xc- (erastin, imidazole ketone

[1]Department of Neurological Surgery, Columbia University Medical Center, New York, NY, USA. [2]Department of Pathology and Cell Biology, Columbia University Medical Center, New York, NY, USA. [3]Department of Chemistry, Columbia University, New York, NY, USA. [4]Department of Proteomics and Macromolecular Crystallography, Columbia University Medical Center, New York, NY, USA. [5]Department of Radiation Oncology, Columbia University Medical Center, New York, NY, USA. [6]Department of Biological Sciences, Columbia University, New York, NY, USA. [7]Department of Medicine, Columbia University Medical Center, New York, NY, USA. [8]These authors contributed equally: Pavan S. Upadhyayula, Dominique M. Higgins, Angeliki Mela. ✉e-mail: pc561@cumc.columbia.edu

erastin (IKE), sulfasalazine)[7,9], compounds that directly inhibit GPX4 (RSL3, ML-210)[8], and compounds that inhibit glutathione synthesis (buthionine sulfoximine)[8]. Cysteine deprivation can also induce ferroptosis, but this effect is mitigated by the transsulfuration pathway[10,11].

Our group has recently shown that a broad range of cancer cell lines, including glioma, is sensitive to ferroptosis inducers[12]. Moreover, we have also demonstrated that compounds that target system Xc⁻ can synergize with radiation to increase reactive oxygen species (ROS) generation and lipid peroxidation in ex vivo organotypic glioma slices[7,13]. We hypothesized that given the centrality of glutathione to protect from ferroptosis, depletion of its precursors, cysteine and methionine, should sensitize cells to undergo ferroptosis. Pharmacologic means of cysteine deprivation have been shown to be efficacious in other cancers[14]. However, blood-brain barrier penetration remains an obstacle for any central nervous system target. We therefore sought to determine the effect of dietary restriction of cysteine and methionine on glioma. We found that cell death, lipid peroxide generation, and transcriptional hallmarks of ferroptosis are enhanced by cysteine and methionine deprivation (CMD). We also found that treating mice with a methionine-restricted cysteine-depleted diet is safe and decreased glutathione levels in vivo. Finally, we found this in vivo dietary paradigm improves therapeutic response to RSL3 and survival in an orthotopic syngeneic murine model of glioma and alters the lipid composition of tumors to create a pro-ferroptotic environment. These results support using CMD diet as a non-invasive method for improving the efficacy of ferroptotic treatments and survival of glioma patients.

## Results

### CMD sensitizes glioma cells to ferroptosis induction

We first characterized the effects of CMD on glioma responsiveness to ferroptosis. Given that cysteine and methionine are critical for the synthesis of glutathione, the key substrate used by the enzyme GPX4 for detoxification of lipid peroxides, we hypothesized that CMD would synergize with GPX4-mediated ferroptosis induction. To this end, basal media was made either with normal DMEM or DMEM without cysteine and methionine. This process is fully described in the methods section and was adapted for cell culture based on our previous ferroptosis permissive glioma culture methods[13,15]. We surveyed the responsiveness of human and murine glioma cell lines (Supplementary Table 1) to ferroptosis induction in the presence and absence of cysteine/methionine. The five glioma cell lines assayed had baseline sensitivity to ferroptosis by the GPX4 inhibitor RSL3 (Fig. 1a), confirmed by live-cell confocal microscopy showing RSL3 mediated induction of lipid peroxidation as evidenced by green fluorescence shift in the Bodipy-C11 dye following addition of RSL3 (Supplementary Fig. 1a). Ferrostatin, a ferroptosis inhibitor, prevented this lipid peroxidation (Supplementary Fig. 1b). RSL3 mediated cell death, however, was not rescuable by necroptosis inhibitors (Nec-1s) or apoptosis inhibitors (ZVAD-FMK) (Supplementary Fig. 1c). Dose response assays demonstrated that RSL3 and ML-210, another GPX4 inhibitor, both had synergistic enhancement of ferroptosis with CMD (Fig. 1b, Supplementary Fig. 1d). Increased sensitivity to RSL3 mediated ferroptosis by CMD was seen in all responsive murine and human glioma cell lines (Fig. 1c, Supplementary Fig. 1e–g). Importantly a low dose (100 nM) of RSL3 in combination with CMD increased lipid peroxidation to levels equivalent to a higher dose of RSL3 (500 nM). The combined effect of CMD plus RSL3 was markedly synergistic as defined by the Chou-Talaly method for quantification of synergy, termed the coefficient of drug interaction (CDI). CDI values <0.7 indicate strong synergy; most cell lines had CDI values <0.1 (Supplementary Fig. 1h). Pre-treatment incubation of glioma cells for 6 h in CMD sensitized tumor cells to subthreshold doses of RSL3 across all murine glioma cell lines as

determined by Bodipy-C11 fluorescence shift (Fig. 1d, e, Supplementary Fig. 1i). Importantly, primary human astrocyte cultures were not sensitive to RSL3 mediated cell death, nor RSL3 + CMD mediated cell death demonstrating a tumor cell specific mechanism of synergistic cell death (Supplementary Fig. 1j).

We next used an ex vivo organotypic slice culture model from a human primary glioblastoma to further validate the effects of CMD[16]. The slices were treated for 24 h with RSL3 and/or CMD and assayed via flow cytometry for levels of reactive oxygen species (ROS) using H2DCFDA. Similar to the in vitro results for lipid peroxidation, a low subthreshold dose of RSL3 (100 nM) plus CMD increased ROS to levels equivalent to a high dose of RSL3 (500 nM) (Control −7.8%, 100 nM RSL3 −2.58%, 500 nM RSL3 52.3%, CMD control −29.7%, CMD + 100 nM RSL3 −53.5%; Fig. 1f). In the primary ex vivo samples, CMD alone was sufficient to increase ROS levels.

### CMD induces transcriptional changes canonically associated with ferroptosis

We then investigated the transcriptional hallmarks of cellular response to CMD. Previous studies have shown that CHAC1, PTGS2, and SLC7a11 mRNAs are upregulated following ferroptotic induction. Furthermore, ATF4 has been tied to amino acid deprivation and ferroptotic stress response as a mechanism to increase SLC7a11 expression and cysteine import[17–19]. We harvested mRNA following 24 h of CMD in the murine glioma cells and 48 h of CMD in the human glioma cells. RT-qPCR of the murine glioma cells showed that by 24 h there were significant increases in CHAC1, PTGS2, SLC7a11 and ATF4 transcripts (Fig. 2a i.–iv.). In the human glioma cells a significant upregulation of CHAC1, SLC7a11 and ATF4 transcripts were seen at 48 h (Fig. 2b i.–iii.). These changes were also seen in the ex vivo setting, where organotypic slices were generated from a post-treatment recurrent GBM (Fig. 2c) and a high-grade R132H IDH1 mutated glioma (Fig. 2d) with neighboring slices being placed into either control media or CMD media. After 24 h, RNA was harvested and RT-qPCR showed significant increases in CHAC1 (Fig. 2c, d i.). The IDH1- mutated glioma had significantly increased SLC7a11 expression following CMD, while the IDH1-wild-type glioma trended towards an increase of SLC7a11 ($p = 0.08$) (Fig. 2c, d ii.). These findings show that CMD induces transcriptional alterations similar to ferroptotic stresses in murine and human gliomas in in vitro and ex vivo settings.

### CMD metabolic alterations in vitro

To characterize further the effects of CMD on glioma cells, we performed targeted metabolite profiling on over 100 metabolites for two murine cell lines (MG1, MG3) treated with CMD for 24 h. Principal component analysis of CMD treated versus control samples demonstrated clear clustering of metabolites according to treatment condition (Supplementary Fig. 2a). An enrichment ratio based upon the number of differentially assessed metabolites within specific metabolite pathways showed that cysteine/methionine metabolism, glycine-serine-proline metabolism, taurine/hypotaurine metabolism, alanine/aspartate/glutamate metabolism and selenocompound metabolism were significantly impacted by CMD (FDR-corrected $p$-value <0.05) (Supplementary Fig. 2b). The heatmap of the top 50 differentially assessed metabolites showed clear separation between CMD and control samples (Supplementary Fig. 2c). This metabolite survey showed a reduction in both oxidized and reduced glutathione (LFC 0.124; FDR-corrected $p$-value <0.05) (Supplementary Fig. 2c). This was further confirmed by a colorimetric assay demonstrating a significant reduction in reduced glutathione across all cell lines with 24 h of CMD (Fig. 3a). To confirm this metabolite survey, a follow-up targeted metabolite experiment with 23 internal standards for metabolites of interest in the cysteine methionine metabolism was performed on MG3 cells. Effects of CMD at 24 and 48 h deprivation were examined. Significant reductions in concentrations of hypotaurine, methionine

and glutamate were noted at 24 h (Fig. 3b); at 48 h both oxidized and reduced glutathione, glutamate, and hypotaurine/taurine were also decreased (Fig. 3d). Significant increases in several amino acids and metabolites were seen at both 24 and 48 h (Fig. 3c, e).

Previous studies have shown that acute oxidative stress can oxidize cysteine residues on proteins necessary for the electron transport chain and citric acid cycle[20]. Thus we hypothesized that CMD would dampen cellular metabolism. Using a mitochondrial stress assay with the Seahorse Analyzer on the murine glioma cells, we measured basal oxygen consumption followed by sequential measurements of ATP-production (oligomycin inhibition), maximal respiration (FCCP inhibition) and mitochondrial respiration (rotenone/antimycin inhibition) (Fig. 3f). Basal respiration, maximal respiration, ATP-linked respiration and proton-leak were all significantly reduced with CMD (Fig. 3g). Importantly, the extracellular acidification rate also decreased,

showing a dampening of both aerobic and anaerobic respiration (Fig. 3h), supporting a global effect of CMD on glioma cell metabolism.

## CMD leads to increased survival in vivo

We next tested the effects of dietary CMD on survival in an orthotopic syngeneic murine glioma model. MG3 cells were orthotopically injected into mice. At 7 days post injection (DPI) the mice were either switched to a control diet (0.43% w/w methionine, 0.40% w/w cystine) or a CMD diet (0.15% w/w methionine, 0% w/w cystine) and maintained on these diets for the remainder of the experiment (Fig. 4a). The diet was tolerated throughout the duration of the survival studies, although notably CMD mice maintained lower weights than control mice (Supplementary Fig. 2d). Kaplan-Meier survival analysis showed a significant survival benefit for CMD mice over control mice, despite the lower weights of the CMD mice (Median Survival: control-40 DPI, CMD-

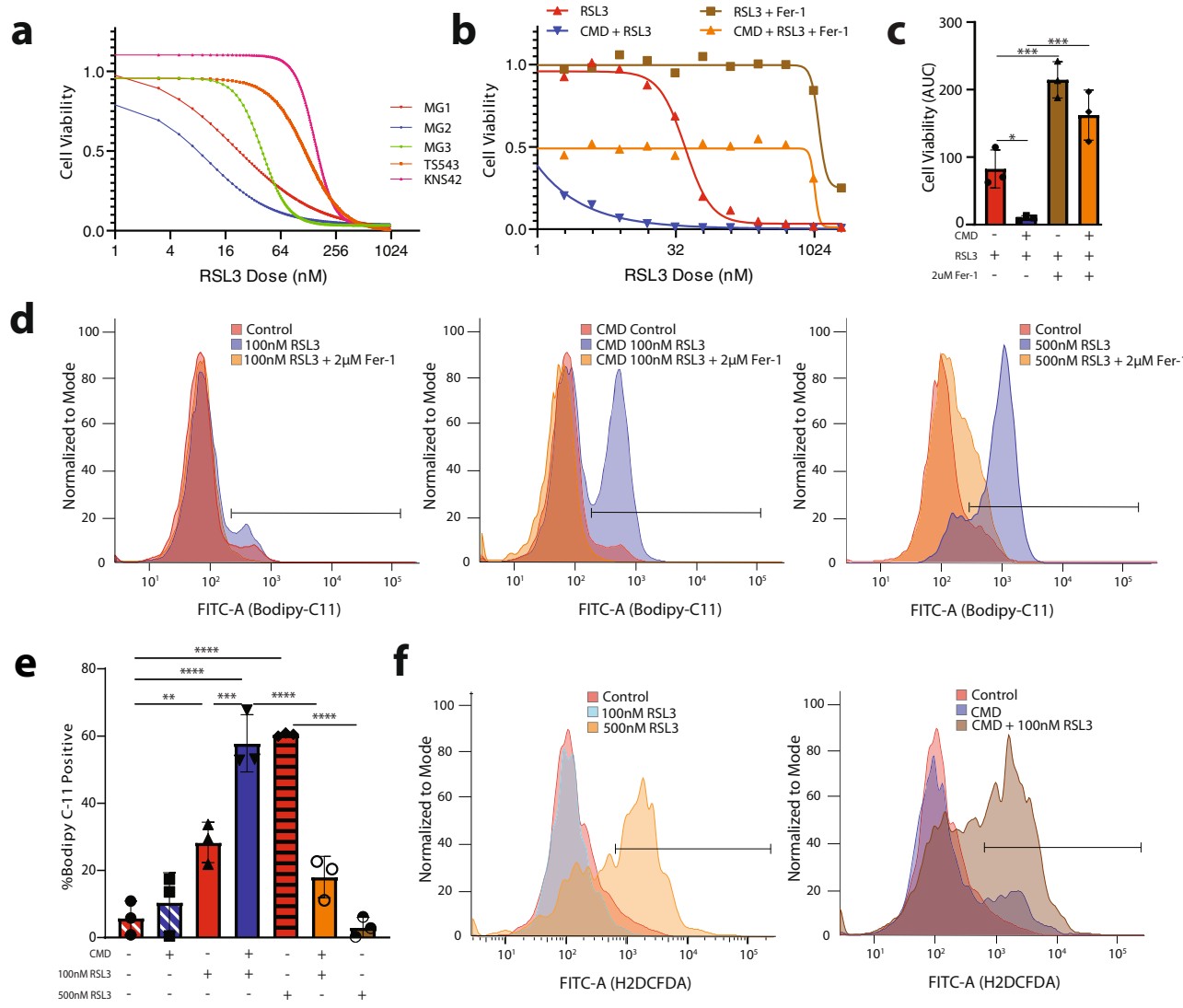

**Fig. 1 | Cysteine and methionine deprivation (CMD) sensitizes glioma to RSL3-induced ferroptosis. a** 384-well dose-response curves showing response to RSL3 from 5 glioma cell lines: MG1, MG2, MG3, TS543, and KNS42. **b** Representative 384-well dose-response showing MG3 cells treated with RSL3 (red), RSL3 plus 2 uM Ferrostatin-1 (brown), CMD plus RSL3 (blue), CMD plus RSL3 and 2 uM Ferrostatin-1 (orange). **c** AUC quantification for dose response curves from 3-independent 96-well dose response curves of MG3 murine glioma cell lines treated with RSL3 ± CMD ± 2 uM Ferrostatin-1. **d** Representative Bodipy-C11 flow data from MG1 cells: left panel shows DMSO control (red), 100 nM RSL3 (blue), and 100 nM RSL3 plus

2 uM Ferrostatin-1 (orange) treatment for 30 min. Middle panel shows the same conditions but with 6 h of cysteine methionine deprivation pretreatment. Right panel shows a higher dose of RSL3 treatment (500 nM). **e** Quantification of 3 independent experiments demonstrated in **d**. **f** Flow cytometry, using H2DCFDA of ex vivo organotypic slice cultures from a human primary glioblastoma (CUMC TumorBank 6193) cultured in control or CMD media and treated with RSL3. Data for **c** and **e** are presented as mean ± SD. Significance denoted by: *$p < 0.05$, **$p < 0.01$, ***$p < 0.001$.

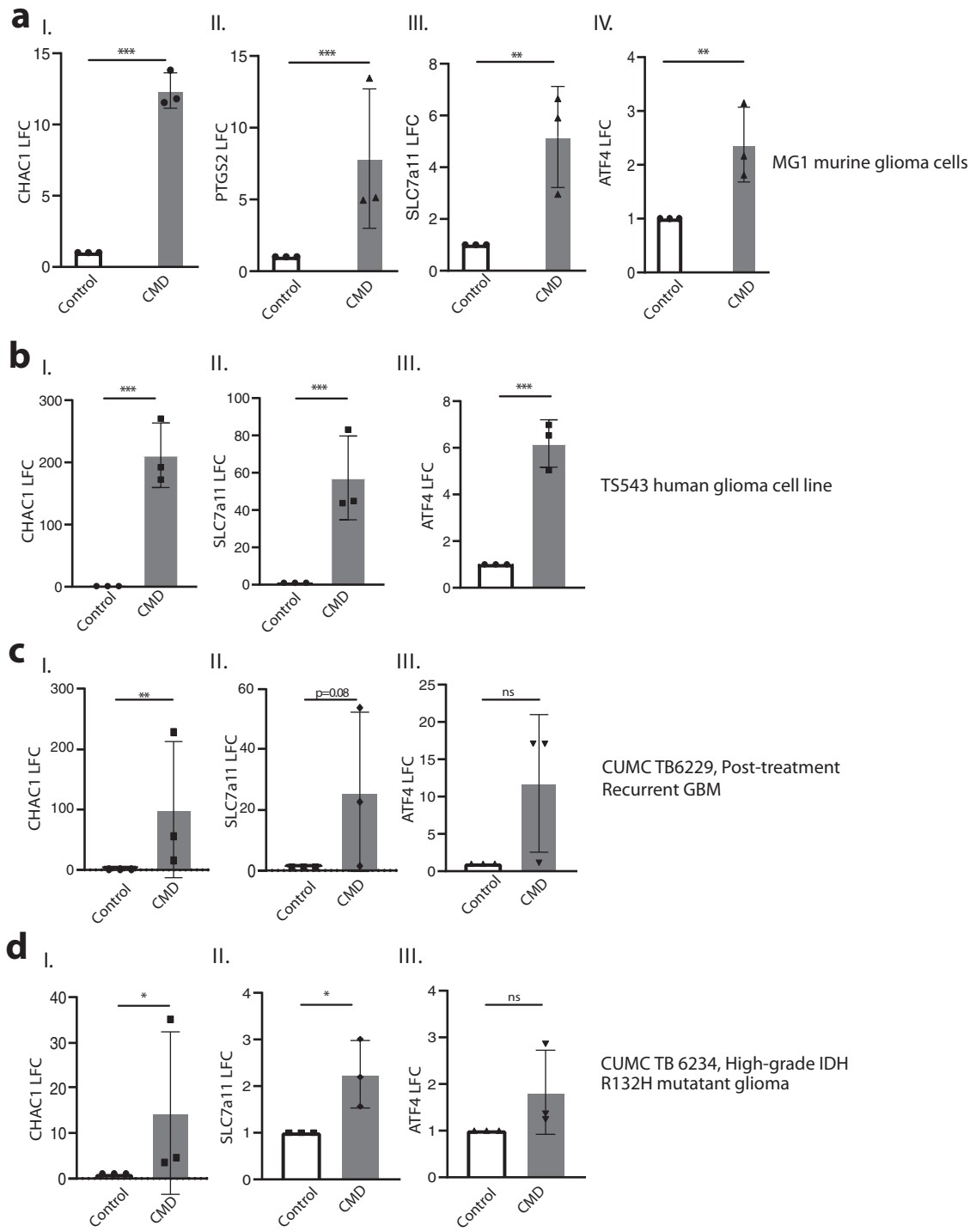

**Fig. 2 | Cysteine methionine deprivation induces transcriptional hallmarks of ferroptosis. a** RT-qPCR data for (I.) CHAC1 ($p < 0.0001$) (II.) PTGS2 ($p = 0.004$) (III.) SLC7a11 ($p = 0.003$) (IV.) ATF4 ($p = 0.007$) transcripts from MG1 cells in either control (black) or 24 h CMD (grey) conditions. **b** RT-qPCR data for TS543 cells after 48 h CMD (grey) compared to control (black) for (I.) CHAC1 ($p < 0.0001$) (II.) SLC7A11 ($p < 0.0001$) and (III.) ATF4 transcripts ($p < 0.0001$). **c** RT-qPCR data of ex vivo organotypic slices for CUMC Tumor Bank 6229 Post-treatment recurrent glioblastoma treated in control (black) or CMD (gray) media. Transcripts for (I.) CHAC1 ($p = 0.006$) (II.) SLC7a11 ($p = 0.08$) and (III.) ATF4 ($p = 0.10$) shown. **d** RT-qPCR data of ex vivo organotypic slices for high-grade R132H mutant glioma, CUMC Tumor Bank 6234 ex vivo organotypic slices in control or CMD media. Transcripts for (I.) CHAC1 ($p = 0.04$), (II.) SLC7a11 ($p = 0.01$), and (III.) ATF4 ($p = 0.11$) shown. Data plotted as mean of log fold change ± SEM, $n = 3$ independent experiments for **a**, **b** and three independent slices for **c**, **d**. Statistics assessed using $t$-test on the un-transformed dCT values. Significance denoted by: *$p < 0.05$, **$p < 0.01$, ***$p < 0.001$.

48DPI; $p = 0.048$) (Fig. 4b). A total of 38 mice (18 control mice – 8 male, 10 female; and 20 CMD mice – 10 male, 10 female) were included.

We next tested the synergistic effects of CMD diet on RSL3 treatment in vivo. MG3 cells were again orthotopically injected into mice; mice were transitioned to a CMD diet or control diet at 7 DPI, and maintained on these diets for the remainder of the experiment. At 21 DPI, convection enhanced delivery pumps were implanted to allow for local delivery of either 500 nM RSL3 or

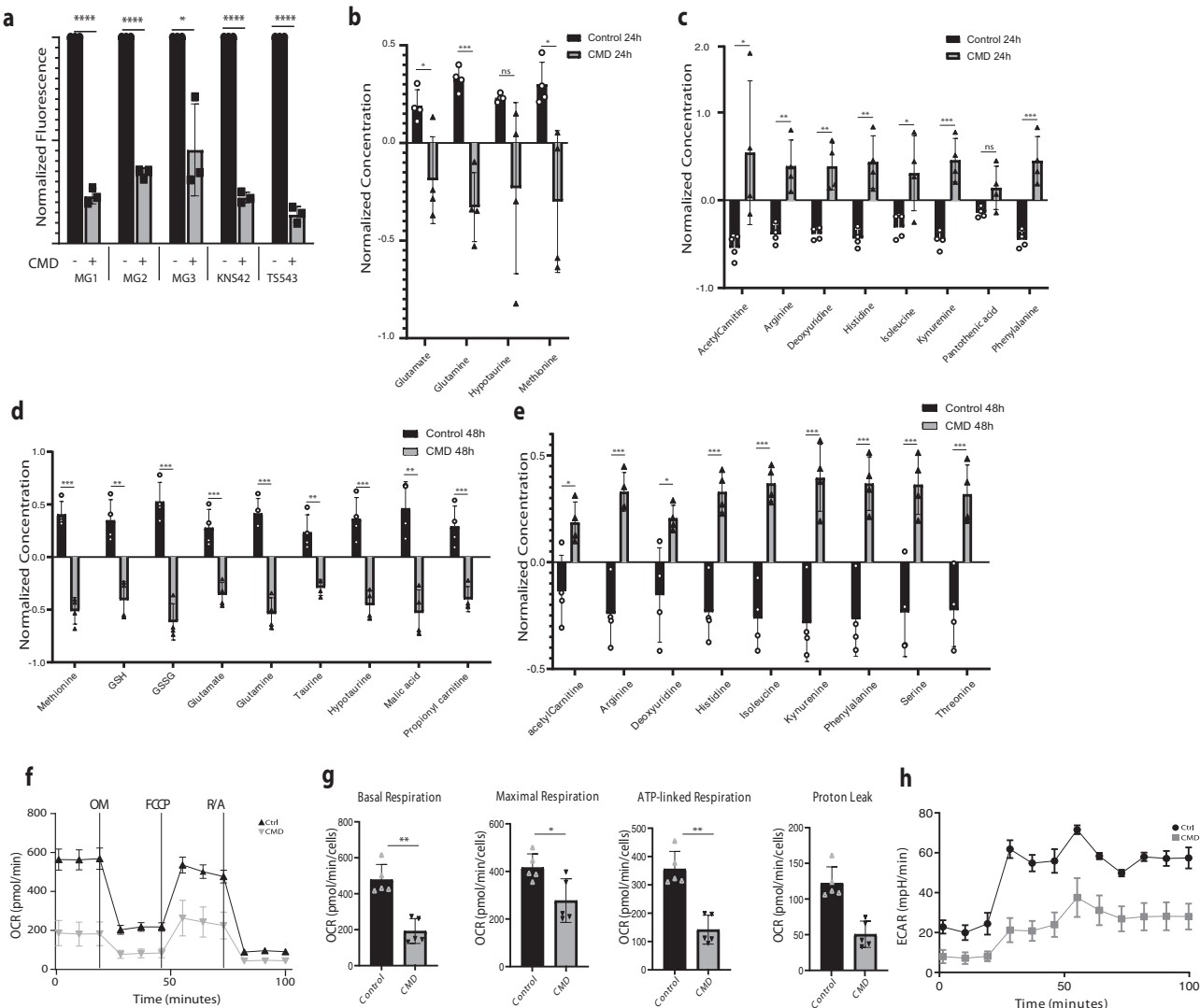

**Fig. 3 | CMD alters glioma cell metabolism.** Metabolite profiling performed on in vitro cell lines (**a**) Colorimetric assay of reduced glutathione levels for (left to right) MG1 ($p < 0.001$), MG2 ($p < 0.001$), MG3 ($p = 0.01$), TS543 ($p < 0.001$), and KNS42 ($p < 0.001$) in control (black bars) and CMD treated cells after 24 h (gray bars), with 3 independent samples per group per cell line. Data is presented as mean ± SD. **b** Normalized metabolite concentrations for key metabolites validated with the standards downregulated in CMD versus control at 24 h, or (**c**) upregulated in CMD versus control at 24 h. **d** Normalized metabolite concentrations for key metabolites with internal standards downregulated in CMD versus control at 48 h, or (**e**) upregulated in CMD versus control at 48 h. **b–e** Log normalized values are

shown, data for each condition is from 4 biological replicates. Statistics are assessed using two tail *t*-tests. Values are presented as mean ± SD. **f–h** Seahorse Mitochondrial stress test of MG3 cells in either control (black) or 12 h CMD (gray) OM: oligomycin, FCCP: Carbonyl cyanide-4 (trifluoromethoxy) phenylhydrazone, R/A: rotenone and antimycin A ($n = 5$). **f** The basal respiration, maximal respiration, ATP-linked respiration and proton leak values calculated from experiment in **g** were calculated and normalized ($n = 5$ per group). Data is presented as mean ± SD. **h** Extracellular acidification rate for control (black) or 12 h CMD (gray). Significance denoted by: *$p < 0.05$, **$p < 0.01$, ***$p < 0.001$.

vehicle control (0.5% DMSO) directly into the tumor volume. Pumps were then explanted after 7 days of infusion (DPI 28) (Fig. 4c). Importantly, mice with the co-treatment of RSL3 and CMD survived significantly longer than control (Median Survival: control − 56 DPI, CMD − 65 DPI, control + RSL3 CED − 64 DPI, CMD + RSL3 CED − 112 DPI; *p*-value control vs. CMD + RSL3 CED = 0.01) (Fig. 4d).

### CMD induces changes within tumor lipidome, metabolome and proteome

We performed targeted metabolite profiling on tumors after acute (2–4 days) and chronic (until end-stage) diet exposure. Targeted metabolic profiling−using internal standards for 23 metabolites of interest similar to the in vitro cells−was performed on MG3 tumors. Mice were transitioned to the diets at 28 days post injection and

tumors were isolated from the anterior frontal region of mice 2 days ($n = 5$, female) and 4 days ($n = 4$, female) after CMD diet initiation along with control mice ($n = 4$, female). Multiple metabolites including acetyl-methionine, adenosyl-homocysteine, cysteine, oxidized glutathione and methionine had decreased concentrations in the tumor volume following the CMD diet. Methionine trended towards a decrease at 2 days with a significant decrease in levels at 4 days (normalized mean control: 0.39; CMD 2 day: −0.017; CMD 4 day: −0.37; CMD 2 day v. control $p = 0.08$; CMD 4 day vs. control = 0.02). Interestingly, although oxidized glutathione showed a decrease at 2 days of CMD, at 4 days this difference was not significant (normalized mean control: 0.24; CMD 2 day: −0.12; CMD 4 day: −0.08; CMD 2 day v. control $p = 0.04$; CMD 4 day vs. control $p = 0.18$). Multiple other metabolites including acetyl-methionine (normalized mean control: 0.37; CMD 2 day: −0.20; CMD 4 day: −0.11; CMD 2 day v. control

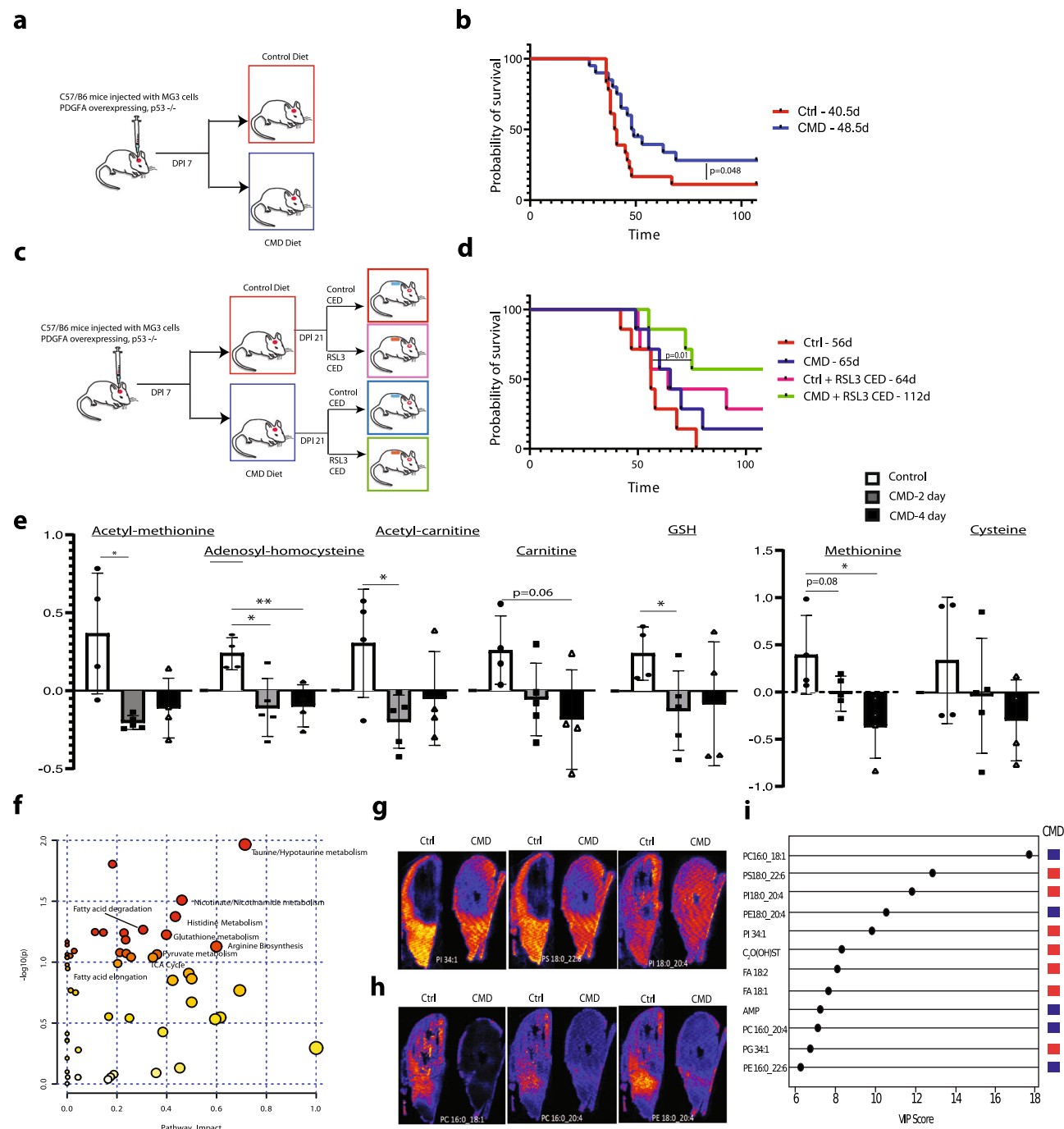

**Fig. 4 | CMD improves survival with corresponding metabolomic, proteomic and lipidomic changes in vivo. a** Diagram of experimental paradigm for CMD in vivo experiments. **b** Kaplan-Meier curve outlining survival comparing control (red, $n = 18$; 8 male, 10 female) versus CMD (blue, $n = 20$; 10 male, 10 female) diet mice orthotopically injected with MG3 cells (Median Survival: Control − 40 days, CMD − 48 days; Gehan-Breslow-Wilcoxon test, $p = 0.048$, Mantel-Cox test $p = 0.051$). **c** Diagram of experimental paradigm for CMD +/− RSL3 experiments. **d** Kaplan-Meier curve outlining survival comparing control ($n = 7$; red, female, median survival − 56d), CMD ($n = 7$, blue, female, median survival − 65d), Ctrl + RSL3 ($n = 7$, pink, female, median survival − 64d), CMD + RSL3 ($n = 7$, green, female, median survival − 112d); (control vs CMD + RSL3, Mantel-Cox $p = 0.010$, Gehan-Breslow Wilcoxon $p = 0.019$). **e** Targeted metabolite profiling of acutely treated in vivo tumors for 23 metabolites in mice transitioned to CMD diet at 28 days post

injection. Tumors were harvested at 2 days after CMD diet initiation (gray), 4 days after CMD diet initiation (black) and from control diet (white) (control $n = 4$, 2-day CMD $n = 5$, 4-day CMD n = 4). Significant metabolites shown as mean ± SEM. Data is presented as mean ± SD. **f** Pathway analysis of metabolite profiling across chronically treated end-stage control ($n = 4$) and CMD ($n = 5$) male mice spanning 200 metabolites with relative concentrations log transformed and samples scaled by mean. Pathways enriched with $p < 0.05$ on one-way labeled. **g** Representative DESI-MS images from tumor region overlay included for upregulated lipid species. **h** Representative DESI-MS images from tumor region overlaid included for down-regulated lipid species. **i** Variable importance of projection shows lipid species important in discriminating the two classes of samples apart (FDR-corrected $p$-value <0.05) from 6 male mice (control $n = 3$, CMD $n = 3$) in the negative ionization mode. Significance denoted by: *$p < 0.05$, **$p < 0.01$, ***$p < 0.001$.

$p = 0.01$; CMD 4 day vs. control $p = 0.07$), acetyl-carnitine (normalized mean control: 0.30; CMD 2 day: −0.20; CMD 4 day: −0.05; CMD 2 day v. control $p = 0.02$; CMD 4 day v. control $p = 0.17$) and propionyl-carnitine (normalized mean control: 0.29; CMD 2 day: −0.21; CMD 4 day: −0.02; CMD 2 day v. control $p = 0.01$; CMD 4 day vs. control $p = 0.23$). These findings point to possible in vivo compensatory mechanisms (Fig. 4e).

Similarly, we performed targeted metabolite profiling on flash frozen tumor tissue harvested from end-stage MG3 tumor bearing mice (control $n = 4$ male mice, CMD $n = 5$ male mice) examining ~200 metabolites of interest. Metabolite profiling of chronic diet end-stage tumors from mice transitioned to the CMD diet or control diet at day post injection 7 showed decreased levels of hypotaurine (FC = 0.12, $p = 0.002$) and oxidized/total glutathione (FC = 0.36, $p = 0.006$), and a trend towards decreased cystathionine (FC = 0.36, $P = 0.08$). (Supplementary Fig. 2e). Quantitative metabolite pathway analysis showed key changes within taurine/hypotaurine metabolism, glutathione metabolism, arginine metabolism, the TCA cycle, and fatty acid elongation/degradation ($p < 0.10$) (Fig. 4f).

To explore changes induced by CMD on glioma cell phenotype at the molecular level, proteomic analysis of adjacent sections from MG3 control ($n = 3$, male) and CMD ($n = 4$, male) mice was performed. CMD induced alterations in numerous protein species (299 protein species differentially expressed; FDR-corrected $p$-value 0.2, |LFC| > 0.58). Of the pathways activated in CMD versus control, the one with the greatest enrichment score was lipid catabolic processes (Supplementary Fig. 2g). Other gene/protein sets enriched in CMD versus control included oxidation/reduction processes, positive regulation of lipid catabolism and cell substrate adhesion and extracellular space. Interestingly, CMD led to a robust immunosuppressive signature involving downregulation of proteins related to antigen presentation and lymphocyte activation (Supplementary Fig. 2g). These findings suggest that CMD could affect patients' response to immunotherapy.

We then performed a joint pathway analysis combining the proteomic differential expression matrix with FDR < 0.2 and |LFC| > 0.58 and the metabolomic differential expression matrix of compounds with |LFC > 0.58| yielding a comprehensive tissue level view of pathways altered by the diet (Supplementary Fig. 2f). Cysteine/methionine metabolism, glutathione metabolism, ferroptosis, glycerophospholipid metabolism were relevant pathways significantly altered based on the joint proteomic and metabolomic analysis. Thus, on both a metabolite and protein level it is clear that CMD led to profound alteration of the tumor microenvironment.

To identify spatial distributions of metabolites and lipids in the tumor tissues and to further analyze metabolic changes induced by ferroptosis, we performed mass spectrometry imaging experiments. Desorption electrospray ionization imaging mass spectrometry (DESI-IMS) was carried out on six end-stage MG3 tumor samples (Control $n = 3$, CMD $n = 3$) in both positive and negative ion modes. Overall, the major detected ions in mouse glioma tissues were lipids including saturated and unsaturated free fatty acids (FFA), phosphatidylcholines (PC), phosphatidylethanolamines (PE), and phosphatidylinositols (PI), phosphatidylserines (PS), phosphatidylglycerols (PG) and sulfatides (ST). The results outlined are from the negative ion mode where more significant changes in lipid abundances were observed between control and CMD groups. The abundance distribution maps for a CMD and a control slice with tumor areas outlined are shown for lipid species increased in CMD (Fig. 4g) and lipid species decreased in CMD (Fig. 4h). Variable importance of projection plots for significantly altered lipid species are shown in Fig. 4i. Among the identified lipid species, the relative abundance of several lipid species including FA 18:2, FA 18:1, PS 18:22:6, PI 18:0_20:4, PI 34:1, PG 34:1, and C20(OH)ST were significantly increased in the tumor regions of the CMD group compared to the control group. In contrast, the relative abundances of PC 16:0_18:1, PE 18:0_20:4, PC 16:0_20:4, and PE 16:0_22:6, and

adenosine monophosphate (AMP) were significantly decreased (FDR-corrected $p$-value <0.05) in the tumor regions of CMD mice compared to the tumors of control mice (Fig. 4i). Lipidomic DESI analysis of non-tumor regions showed that the majority of lipid profiles in the non-tumor regions were not significantly altered, pointing to tumor specific effects of the CMD diet (Supplementary Table 2).

To examine transcriptional changes in vivo, we injected MG3 cells in mice that were then treated with the four conditions (control diet + DMSO local delivery, CMD diet + DMSO local delivery, control diet + RSL3 local delivery and CMD diet + RSL3 local delivery), and harvested the tumors after 3 days of treatment. RT-qPCR was performed on tissue lysates to examine transcriptional changes associated with each condition. To account for treatment related decreases in tumor cells, we normalized transcript expression to olig2 levels within samples, as our previous studies have shown that olig2 is highly expressed by tumor cells in our murine glioma model[21]. Our data show that co-treatment with CMD diet and RSL3 significantly upregulated transcripts canonically associated with ferroptosis in vivo (Supplementary Fig. 3a–c).

## Discussion

Ferroptosis is a promising avenue for cancer treatment[22,23]. This novel form of programmed cell death is induced by inhibition of GPX4, an enzyme that facilitates glutathione mediated detoxification of toxic lipid peroxides[24]. Herein, we demonstrate that murine and human glioma cells are susceptible to ferroptosis via GPX4 inhibition by drugs such as RSL3. We also show that RSL3 mediated cell death is ferroptosis specific (independent of apoptosis or necroptosis) and is associated with increased lipid peroxidation. Moreover, we demonstrate that nutrient deprivation of cysteine and methionine decreases cell survival, and synergistically increases lipid peroxidation and cell death when combined with RSL3. Ex vivo slices from human gliomas showed both synergistic sensitivity to CMD and ferroptosis inducers as well as significant transcriptional upregulation of CHAC1 and SLC7a11 following CMD. Notably, in vivo dietary deprivation of cysteine and methionine resulted in increased survival, though modest, with distinctive changes in the lipidomics, proteomics and metabolomic profile of the tumors. Co-treatment with CMD diet and local delivery of RSL3 led to a robust survival improvement over control conditions.

Our analysis of the effects of CMD in vitro showed significant decreases in metabolites in 3 major pathways including cysteine-methionine metabolism, taurine/hypotaurine metabolism, glutathione synthesis. These findings translated to the in vivo setting where an orthotopic mouse glioma model treated with CMD diet showed decreases in pathways related to glutathione synthesis, and hypotaurine/taurine metabolism, demonstrating that systemic dietary deprivation affects tumor metabolism and growth within the central nervous system. Previous studies have shown that depleting glutathione with the drug Erastin leads to ferroptosis[8]. We show that CMD can decrease glutathione in glioma cells in both the in vitro and in vivo settings.

We also show the CMD diet is a ferroptotic stress as demonstrated through a multi-omic approach. Our analysis showed that even within 2-4 days metabolite profiles of tumors showed reductions in methionine and glutathione, among other metabolites. End-stage tumors showed alterations in a host of lipids and metabolites associated with oxidative stress and aerobic/anaerobic respiration. DESI-IMS data showed a shift in tumor lipid profiles towards more pro-ferroptotic species. The levels of PI 38:4 and PS 40:6 (PI 18:0_20:4, PS 18:22:6), phospholipids with PUFA tails, were increased significantly in the CMD group. Moreover, it has been shown that phospholipids with saturated and monounsaturated fatty acid tails are ferroptosis resistant. PC 16:0_18:1, one of the most abundant phospholipids in the brain with proven anti-ferroptotic activity[25], was depleted significantly in the CMD group. Many of the other compounds significantly decreased in

CMD tumors (PE 18:0_20:4, PC 16:0_20:4, and PE 16:0_22:6) have been previously reported as depleted after treatment with a known ferroptosis inducer, imidazole ketone erastin[26]. Notably we saw upregulation of FA18:2, a omega-6 PUFA tied to decreased antioxidant capacity in vivo[27,28].

Previous studies have looked at ketogenic diets, caloric restriction and high-fat low carb diets in the CT-2A orthotopic murine astrocytoma models[29–34]. Here we show that CMD diet was tolerated throughout the duration of the survival studies and was associated with a significant survival benefit, indicating local effects on brain tumor growth and response to RSL3. Notably, the mice on the CMD diet maintained a significantly lower weight, and there may be issues with systemic toxicity that will need to be addressed in future studies. In identifying a transcriptional and metabolic signature we hope to aid future translational efforts with the CMD diet by providing a profile to assess treatment efficacy. The CMD diet was also associated with key tumor specific metabolic and lipid changes that are promising avenues for future investigation and combination treatment. Notably, the tumor microenvironment is heterogenous, and regional variance associated with necrosis, microvascular proliferation and infiltrated brain tissue may contribute to the patterns of phospholipid abundance. Future studies are needed to determine the effects of CMD on the brain tumor microenvironment and to assess whether they are therapeutically actionable. Strategies combining the sensitizing CMD diet with either classic ferroptosis inducing agents, RSL3, and non-canonical inducers like radiation hold additional therapeutic potential. Our data adds to a growing body of evidence showing that certain cancers, including glioma, are sensitive to ferroptosis and rely on cysteine metabolism to deal with oxidative and ferroptotic stress[14,35].

## Methods

### Ethics statement
All experimental procedures involving mice were reviewed and approved by the Columbia University Institutional Animal Care and Use Committee (IACUC). The mice were monitored daily and euthanized when they exhibited signs that demonstrate that they are symptomatic from tumor burden including any of the following; periorbital hemorrhages, epistaxis (nose bleeds), seizures, decreased level of alertness and activity, impaired motor function, and/or impaired ability to feed secondary to decreased motor function, paresis or coma. Procedures involving de-identified human surgical specimens were conducted following Columbia University Irving Medical Center Institutional Review Board approval.

### Cell lines and culture conditions
Murine glioma cell lines were generated from retrovirus induced orthotopic murine glioma models as previously described[21]. Briefly, C57Bl/6 neonatal (p4) mice harboring floxed p53 and stop-flox mCherry-luciferase were anesthetized using hypothermia and orthotopically injected with a PDGFA–internal ribosomal entry site (IRES)–cyclization recombination (Cre) retrovirus (stereotaxic coordinates relative to bregma: 1 mm anterior, 1 mm lateral, 1 mm deep), resulting in tumor cells that overexpress PDGFA and mCherry-Luciferase, and have deleted p53[36]. End-stage tumors were harvested and tumor cells isolated and cultured in basal media (BFP), containing DMEM (Gibco™ 11965092) with 0.5% FBS (Gibco™ 16000044), antibiotic-antimycotic (Thermo Scientific 15240096), N2 supplement (Thermo Fisher Scientific, 17502-048), and 10 ng/ml each of recombinant human PDGF-AA (Peprotech, 100-13 A) and FGFb (Peprotech, 10018B50UG). Three biological replicates of PDGFA driven cells made from three independent tumors with the same genetic background were used for this study. A Pten[−/−] P53[−/−] PDGFB[+] cell line was also used[21]. All cells were grown at 37 °C with 5% CO2. For all murine glioma cell lines cysteine methionine deprived media was made from basal DMEM without cysteine, methionine and glutamine (Thermo Fisher

Scientific, 21013024) that was supplemented with L-glutamine to a final concentration of 4 mM. All other components of the media were the same for control and CMD media. Human glioma cells were cultured as previously described[37,38]. TS543 cell neurosphere cell lines were cultured in Neurocult media as previously described[37], but were dissociated and plated in a single cell monolayer in 96 well plates in either BFP or cysteine methionine deprived BFP for dose response experiments. KNS42 cell lines were cultured in DMEM + 10%FBS or cysteine methionine deprived DMEM + 10%FBS. Thus, for all experiments the only difference between CMD media and control media used for each cell line was the concentration of cysteine and methionine.

### Generation of acute organotypic slice cultures from mouse brains and human surgical specimens
Mouse or human brain slice cultures were generated as described[16]. Mice were sacrificed by cervical dislocation. The brain was removed and placed into an ice-cold sucrose solution (210 mM sucrose, 10 mM glucose, 2.5 mM KCl, 1.25 mM NaH2PO4, 0.5 mM CaCl2, 7 mM MgCl2 and 26 mM NaHCO3). After 20 min, the brain was cut into 300–500 μm sections using a McIlwain Tissue Chopper. After cutting, slices were rested in the sucrose solution for 20 min. Then the tissue sections were transferred onto Millicell cell culture inserts (0.4 μM, 30 mm diameter) and placed in 6-well plates containing 1.5 mL of medium consisting of DMEM/F12 with N-2 Supplement and 1% antimycotic/antibiotic.

Human surgical specimens were collected from Columbia University Medical Center operating theaters, deidentified and placed in a sterile 50 mL conical tube containing the ice-cold sucrose solution for transportation. For treatment conditions, Hams-F12 without cysteine or methionine (MyBioSource, MBS652871) was mixed 1:1 with DMEM without cysteine or methionine (Thermo Fisher Scientific, 21013024) to make the DMEM/F12 without cysteine/methionine. Slices were plated into treatment condition media and incubated for 24 h after which dissociation experiments were conducted.

### Real Time Quantitative Polymerase Chain Reaction Primers
Primers were found using the Harvard qPCR Primer Bank[39]. The primers sequences used are outlined below in Supplementary Table 3.

### Real time Quantitative PCR Method
RNA was extracted using the RNeasy Mini kit (QIAGEN, 74106). For tissue lysis, a 5 mm stainless steel bead (QIAGEN, 69989) was used to facilitate tissue lysis prior to RNA extraction. Following RNA extraction, up to 2.5 μg of RNA was used with the SuperScript Vilo cDNA synthesis kit (ThermoFisher, 11754050). cDNA was diluted to a concentration of 250 ng/μL and the RT-qPCR reactions were conducted with Thermo Scientific ABsolute Blue qPCR SYBR (ThermoFisher, AB4322B). Duplicate samples per condition were analyzed on an Applied Biosystems QuantStudio 3 qPCR instrument with all experiments being repeated 3 independent times. beta-Actin was used as reference and log fold change was calculated using the ddCT method comparing treatments to a control sample.

### Cell viability assays—RSL3
Cell viability was assessed using the Cell-Titer Glo luminescence assay. Murine glioma cells were plated in triplicate at a density of 6000 cells per well in a 96-well plate (ThermoFisher Scientific, 165305). 24 h after plating, media was removed and treatment media was added. Viability was assessed 24 h after treatment. Human glioma cells were plated at a density of 2000 cells per well. Cells were plated in normal media or cysteine methionine deprived media. 24 h after plating, media was changed into drug treatment. 48 h after plating luminescence was quantified. Averages across 3 independent experiments are reported. For experiments conducted in 384-well plates, mouse glioma cells were plated at a density of 1600 cells per well and human glioma cells were plated at a density of 1000 cells per well.

Similarly, Human astrocytes (ScienCell #1800) were plated in triplicate at a density of 6000 cells per well in a 96-well plate. Cells were plated in normal media or cysteine methionine deprived media. 24 h after plating, media was changed into drug treatment. 48 h after plating luminescence was quantified. Averages across 3 independent experiments are reported.

The assays as described above were quantified using Cell-Titer Glo (Promega) ATP based bioluminescence. To determine cell viability, a 50% Cell Titer Glo and 50% cell culture medium was added to each well and incubated at room temperature for 10 min. Luminescence was assessed on a Promega GloMax Microplate Reader.

## Flow cytometric analysis of Lipid Peroxidation or ROS
**Cell lines.** Adherent cells were lifted using TrypLE (ThermoFisher Scientific, 12604013). Cell pellets were resuspended in 1 mL PBS with either Bodipy-C11 (ThermoFisher, D3861) or H2DCFDA (ThermoFisher, D399) were added to a final concentration of 2 uM and 5 uM respectively. Cells were incubated with the dyes for 10 min at 37 degrees Celsius. Cells were centrifuged at $400\,g \times 5$ min then resuspended in PBS.

**Slice Cultures.** Slice cultures were dissociated in Papain (9.5 mL DPBS, 500 μL papain, 1.67 μL 6 M NaOH, 2 mg L-cysteine, 100 μL DNAse) and incubated in a warm bath shaker at 37 °C for 30 min. After centrifugation at 400 g for 5 min, papain was aspirated and slices were resuspended in ice cold PBS and triturated with glass tip pasteur pipettes. This process was repeated 1x and then the cell suspension was resuspended in a 30% sucrose solution and spun at $1000\,g \times 5$ min. The cell suspension was resuspended in PBS and stained with Calcein Blue (final concentration 5 μM) and H2DCFDA (final concentration 10 μM), incubated in a water bath at 37oC for 10 min. Suspensions were spun down at $500\,g$ for 5 min and resuspended in PBS and taken for flow cytometric analysis on a LSRIII Fortessa machine.

Cell and Slice culture suspensions were filtered in polystyrene flow tubes (Fisher Scientific #352008). Data were collected on an LSRIII Fortessa flow analyzer and analyzed using FlowJo v10.

## Time lapse confocal imaging of lipid peroxidation
$1.3 \times 10^5$ MG1 cells were plated on poly-L-lysine coated 35 mm glass-bottom dishes (MatTek life sciences) for 24 h. Cells were incubated for 30 min in BFP media containing 5 μM BODIPY-C11. Media were replaced with fresh BP and cells were imaged on a Nikon A1RMP confocal microscope at 37 °C in a humidified chamber with 5% $CO_2$. Time-lapse images were acquired using a 40×/1.3 NA oil immersion objective and focus was maintained using the Perfect Focus System. Excitation was achieved using 488 nm and 561 nm laser illumination; emission of the oxidized and reduced forms of BODIPY-C11 was captured using a 525/50 and a 595/50 filter, respectively. At time 0, RSL3 (500 nM) or RSL3 (500 nM) + Ferrostatin (2 μM) were added and images were acquired every 30 s for a total of 30 min. Images were exported to ImageJ-FIJI for analysis.

## Extracellular flux analysis and FAO assay
This process using a Seahorse XFe24 analyzer is described in depth elsewhere[40]. A mitochondrial stress assay and fatty acid oxidation assay based of Agilent Technologies manual. Murine glioma cells (MG1 and MG3) were seeded in XFe24 cell culture microplates (Agilent TEchnologies) at 18,000 cells per well in 250 μL of BFP described above. After 4 h, media was aspirated and replaced with either BFP or CMD BFP media. Treatments were continued for 18 h. Mitochondrial stress tests were run with the following concentrations of media: 10 mM glucose, 2 mM glutamine, and 1 mM pyruvate in assay medium, and 2 μM oligomycin, 2 μM trifluoromethoxy carbonylcyanide phenylhydrazone (FCCP), and 0.5 μM rotenone/antimycin A. The assay involved injection of glucose (10 mM), followed by oligomycin (1 μM),

followed by 50 mM 2-deoxy-d-glucose. Fatty acid oxidation assays were run using glucose (0.5 mM), glutamine (1 mM), 0.5 mM l-carnitine and BSA conjugated palmitic acid.

## Animals, orthotopic tumor implantation, and diet allocation
**Cell implantation.** Mice were anesthetized with Ketamine/Xylazine (100 mg/kg and 10 mg/kg, respectively) and assessed for lack of reflexes by toe pinch. Hair was shaved and scalp skin incised. The skull was cleaned with a q-tip and bregma identified. A burr hole was made with a 17 gauge needle 2 mm lateral and 2 mm anterior to the bregma. Cell suspension was made from lifted adherent cell lines. Intracranial injection ($5 \times 10^4$ MG3 cells in 1 μL) performed under stereotactic guidance, 2 mm deep into the brain parenchyma aiming for subcortical white matter, using a Hamilton syringe at a flow rate of 0.25 μL/min. Tumor growth was assessed through monitoring of luciferase signaling by bioluminescence imaging as previously described[41]. Mice were injected at 6 weeks for diet survival studies and 8 weeks for diet and convection enhanced delivery pump studies.

**Diet allocation.** Special diets were created by LabTest Diet (W.F. Fisher and Sons). A normal chow was used as a baseline (catalog no. 5CC7). From this two diets were created for experimental purposes. The diets used were a control diet with a defined 0.43% methionine and 0.33% cystine (w/w, catalog no. 5WVL) and a cystine deprived-methionine restricted diet with 0.15% methionine and 0.0% cystine (w/w, catalog no. 5WVM). Similar diets have shown safety in mouse experiments[42]. Mice were transitioned to the diet seven days post tumor implantation. Investigators were not blinded to the allocation during experiments or outcome assessments.

**Convection enhanced delivery.** Pump implantation was conducted as previously described[41]. After randomization to control or CMD diets, mice were further randomized into control or RSL3 pump groups. Mini osmotic pumps (Alzet, model 2ML1) were filled with either 0.5% DMSO in PBS (control) or 500 nM RSL3. These pumps infuse ~200 μL of fluid over 7 days. Pump implantation was carried out on DPI 21 as previously described[43]. Explant was conducted on DPI 28 and mice were followed until endstage.

## Tissue Collection
Mice were assessed daily for signs of tumor morbidity. Mice were anesthetized with intraperitoneal injection of ketamine/xylazine (100 mg/g and 10 mg/kg, respectively). Following cessation of toe pinch reflex, mice were perfused with PBS. For end stage tumors the anterior most portion of the brain, which encompassed the anterior most tip of tumor was sectioned and placed in 4% PFA. Remaining brains were harvested and placed on an aluminum weigh boat floating in liquid nitrogen for flash freezing. For short time point experiments the whole brain was flash frozen as described.

## Metabolomic Profiling
**Sample preparation in vivo.** Tumor areas were cored out from whole frozen brains and weighed. 80% HPLC grade methanol or water/methanol/acetonitrile (1:4:4; v/v/v) were added in 1.5 mL eppendorf tubes with excised tumors. Tissue was homogenized and incubated at −80 °C then centrifuged at $14,000 \times g$ for 20 min at 4 °C. Supernatant was transferred to a fresh vial and dried using SpeedVac. The dried sample reconstituted in water/acetonitrile (1:1; v/v) before LC-MS analysis.

**Sample Preparation in vitro.** $2 \times 10^6$ MG1 or MG3 cell lines were plated on a 10 cm dish. 24 h after plating cells were switched to control basal media or CMD basal media. 24 h after treatment, plates were washed 2 times with ice-cold PBS. Plates were aspirated, placed on dry ice and 1 mL of 100% HPLC grade methanol was added to the dish. Cells were

scraped and transferred to cold eppendorf tubes. Metabolites were extracted similar to the tissue samples as mentioned above. Protein was extracted from pellets after centrifugation using cell extraction buffer with protease and phosphatase inhibitors. A colorimetric Bradford assay was read at 740 nm for evaluation of protein content.

**LC-MS Data Acquisition and Processing.** LC-MS analyses were performed on a Q Exactive Orbitrap mass spectrometer (Thermo Scientific) coupled to a Vanquish UPLC system (Thermo Scientific). The Q Exactive operated in polarity-switching mode. A Sequant ZIC-HILIC column (2.1 mm i.d. × 150 mm, Merck) was used for separation of metabolites. Flow rate was set at 150 μL/min. Buffers consisted of 100% acetonitrile for mobile B, and 0.1% NH4OH/20 mM CH3COONH4 in water for mobile A. Gradient ran from 85% to 30% B in 20 min followed by a wash with 30% B and re-equilibration at 85% B. Data analysis was done using TraceFinder 4.1 (ThermoFisher Scientific). The second set of targeted metabolomics analysis were performed on a Waters Acuity UPLC I-Class system equipped with a Waters Acuity UPLC BEH Amide column (100 mm × 2.1 mm i.d) couple with Waters Synapt G2-Si mass spectrometer. The oven temperature was maintained at 45 °C, and the flow rate was set at 0.4 mL/min. The chromatographic separations were performed by the following parameters: solvent A consisted of water with 10 mM ammonium formate and 0.125% formic acid, solvent B was made from acetonitrile/water (95/5, v/v) with 10 mM ammonium formate and 0.125% formic acid. A gradient run was set up as 0–1.5 min at 99% B, 1.5-10 min from 99% to 30% B, 10–10.1 min from 30% to 99% B, 10.1–15 min re-equilibrates at 99% B. The Synapt G2-Si mass spectrometer was operated in both positive and negative electrospray ionization (ESI) mode. A capillary voltage and sampling cone voltage of 1.8 kV and 30 V were used. The source and desolvation temperature were kept at 120 °C and 500 °C, respectively. Nitrogen was used as desolvation gas with a flow rate of 800 L/hr. The data was collected in duplicates in the data independent (MSE) mode over the mass range m/z 50 to 650 with an acquisition time of 0.2 s per scan.

Metabolites were identified on the basis of exact mass within 5 ppm and matching the retention times with the standards. Relative metabolite quantitation was performed based on peak area for each metabolite. In vivo samples were normalized by weight and in vitro samples normalized by protein content using a Bradford assay. Data analysis was performed following log normalization and metabolite by metabolite mean subtraction. Metaboanalyst 5.0 (metaboanalyst.ca) was used for principal component analysis, differential assessment analysis, statistical tests, and quantitative pathway analysis[44]. All tissue based comparisons were performed between mice of the same sex(-male mice were used for end-stage long term metabolite profiling, proteomic and lipidomic experiments, female mice were used for short-term metabolomic profiling).

**Global Quantitative Proteomics analysis**
Tissue from mice with MG3 tumors placed on CMD or control diets as previously described was fixed at end-stage in 4% PFA and paraffin embedded. 5 μM sections were made from blocks; tissue cores were scraped off slides and transferred to 1.5 mL eppendorf tubes. Tissue lysis and de-crosslinking was performed as described by Marchione et al.[45]. Briefly tissue was suspended in 50 uL of 5% SDS/300 mM Tris pH 8.5 and sonicated/boiled in a water bath (at 90 °C × 90 min). Samples were centrifuged then sonication/boiling was repeated. (90 °C × 10 min). The de-crosslinked lysate was centrifuged at 16,000 g in a benchtop centrifuge for 10 min and collected in a new Eppendorf tube. Cleared lysate was precipitated using the "salt method" as previously described[46]. Pellets were resuspended in SDC lysis buffer[47](1% SDC, 10 mM TCEP, 40 mM CAA and 100 mM TrisHCl pH 8.5) and boiled for 10 min at 45 °C, 1400 rpm to denature, reduce, and alkylate cysteine, followed by sonication in a water bath. Samples were then cooled down to room temperature. Protein digestion was processed

overnight by adding LysC and trypsin in a 1:50 ratio (μg of enzyme to μg of protein) at 37 °C and 1400 rpm. Peptides were acidified by adding 1% TFA and vortexing followed by StageTip clean-up via SDB-RPS. Peptides were loaded on one 14-gauge StageTip plugs. Peptides were washed two times with 200 μL 1% TFA 99% ethyl acetate followed by 200 μL 0.2% TFA/5%ACN in centrifuge at 3000 rpm, followed by elution with 60 μL of 1% Ammonia, 50% ACN into eppendorf tubes and dried at 60 °C in a SpeedVac centrifuge. Peptides were resuspended in 7 μL of 3% acetonitrile/0.1% formic acid and injected on Thermo Scientific™ Orbitrap Fusion™ Tribrid™ mass spectrometer using the DIA method[48] for peptide MS/MS analysis. The UltiMate 3000 UHPLC system (Thermo Scientific) and EASY-Spray PepMap RSLC C18 50 cm × 75 μm ID column (Thermo Fisher Scientific) coupled with Orbitrap Fusion (Thermo) were used to separate fractionated peptides with a 5–30% acetonitrile gradient in 0.1% formic acid over 120 min at a flow rate of 250 nL/min. After each gradient, the column was washed with 90% buffer B for 5 min and re-equilibrated with 98% buffer A (0.1% formic acid, 100% HPLC-grade water) for 40 min. Survey scans of peptide precursors were performed from 350-1200 m/z at 120 K FWHM resolution (at 200 m/z) with a $1 \times 10^6$ ion count target and a maximum injection time of 60 ms. The instrument was set to run in top speed mode with 3 s cycles for the survey and the MS/MS scans. After a survey scan, 26 m/z DIA segments were acquired from 200-2000 m/z at 60 K FWHM resolution (at 200 m/z) with a $1 \times 10^6$ ion count target and a maximum injection time of 118 ms. HCD fragmentation was applied with 27% collision energy and resulting fragments were detected using the rapid scan rate in the Orbitrap. The spectra were recorded in profile mode. DIA data were analyzed with directDIA 2.0 (Deep learning augmented spectrum-centric DIA analysis) in Spectronaut Pulsar X, a mass spectrometer vendor independent software from Biognosys. The default settings were used for targeted analysis of DIA data in Spectronaut except the decoy generation was set to mutated. The false discovery rate (FDR) will be estimated with the mProphet approach and set to 1% at peptide precursor level and at 1% at protein level. Results obtained from Spectronaut were further analyzed using the Spectronaut statistical package. Significantly changed protein abundance was determined by unpaired t-test with a threshold for significance of $p < 0.20$ (permutation-based FDR correction) and 0.58 log2FC.

**Desorption Electrospray Ionization–Imaging Mass Spectrometry (DESI-IMS)**
**Tissue preparation.** Consecutive coronal brain sections were cut at −20 °C into 12 μm thick sections on a cryostat (Leica, CM3050S), and directly thaw-mounted onto SuperFrost Plus glass Microscope Slides (Fisherbrand, 12-550-15). Before analysis, the sections were dried under vacuum in a desiccator for 15 min. High resolution mass spectrometry with desorption electrospray ionization (DESI) source was used to scan slices. After the DESI-MSI experiment, the tissue sections were stained with Hematoxylin and Eosin (H&E). A clinical pathologist (PC) identified and outlined tumor regions. The identified region was superimposed upon DESI-MSI maps to extract specific quantitative morphometry allowing for statistical comparisons between tumor regions of CMD mice versus control mice.

**DESI-IMS Data Acquisition and Processing.** The tissue sections were imaged at 50 μm resolution on a Prosolia 2D-DESI source mounted on the SYNAPT G2-Si q-ToF ion mobility mass spectrometer. The electrospray solvent consisted of methanol/water/formic acid (98:2:0.01; v/v/v) containing 40 pg/μL of leucine enkephalin as internal lock mass. The flow rate was 2 μL/min. The spray capillary voltage was set to 0.6 kV, the cone voltage was 50 V, and the ion source temperature was set to 150 °C. Mass spectra were acquired using negative ionization mode with the mass range of m/z 50 to 1200. DESI imaging of all tissue samples were run in a randomized order using the same experimental

conditions in duplicates. Ion image mass spectral data (corresponding m/z features in every pixel within the image) from DESI-MSI was processed for visualization using Waters High Definition Imaging (HDI-imaging, V1.5) software. The images were normalized to the total ion current. Group differences were calculated using a two-tailed parametric Welch's *t*-test with a false discovery rate (FDR) of 0.05 or less as significant. The lipid ions were annotated by searching monoisotopic masses against the available online databases such as METLIN and Lipid MAPS with a mass tolerance of 5 ppm and also matching the drift times with the available standards.

## Statistics and Reproducibility statement

No specific calculations or statistical methods were used to predetermine sample size. Studies were performed using technical controls and biological replicates, based on cell line and tissue availability. At least three samples per condition were analyzed in all cases. In vivo experiments included more than 3 mice to obtain tissue for analysis, or in the case of survival studies, at least 8 mice were included per group, based on the number of mice approved under our IACUC protocol. For in vivo studies, mice were randomly assigned to treatment groups, three weeks after tumor cell injections. For in vitro studies, cells were plated at the same time in 96-well plates and then randomized into different treatment groups based on plate row or column number. Investigators were not blinded to group allocation during in vivo experiments, since CMD diet looks different than the regular mouse diet and different groups had to be placed in separate cages. Investigators were not blinded to treatment groups during data collection. However, blinding is not relevant because data collection was done in an automated way using analyzers. No data were excluded from the analyses. All replication attempts were successful.

## Reporting summary

Further information on research design is available in the Nature Portfolio Reporting Summary linked to this article.

## Data availability

The mass spectrometry proteomics data have been deposited to the ProteomeXchange Consortium via the PRIDE partner repository with the dataset identifier PXD039476. The mass spectrometry raw data files and metadata from both targeted metabolomics and mass spectrometry imaging are deposited to the EMBL-EBI MetaboLights database with the identifier MTBLS5959. Source data are provided with this paper.

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

## Acknowledgements

We thank Drs. Xiaoyuan Yang and Guoan Zhang from Proteomics and Metabolomics Core Facility at Weill Cornell Medicine for conducting the LC-MS metabolomics analysis. We thank Dr. Cameron Brennan for contribution and use of the TS543 cell line. We thank Trine Giaever for her artistic help with the figures. This study was supported by Emerson Health Collective Cancer Research Fund to P.C. and B.R.S., National Institute of Neurological Disorders and Stroke (NINDS) grant R01NS103473, National Center for Advancing Translational Sciences grant UL1TR001873, the William Rhodes and Louise Tilzer Rhodes Center for Glioblastoma, the Khatib Foundation, the Gary and Yael Feger Foundation to P.C. and J.N.B., NCI grants P01CA87497 and R35CA209896 and NINDS grant R61NS109407 to B.R.S. These studies used the resources of the Cancer Center Flow Core Facility and the Confocal and Specialized Microscopy Shared Resource, supported by NIH/NCI Cancer Center Support Grant P30CA013696. The A1RMP confocal microscope was purchased with NIH grant S10RR025686.

## Author contributions

P.C., A.D., M.A.B., D.M.H., J.N.B., B.R.S., S.C., A.Me. and P.S.U. conceived and planned all experiments. P.S.U., D.M.H., A.Me., A.D., A.Ma., K.C., L.L., C.K., C.P.S., B.L.S. performed all in vitro experiments. P.S.U., A.Me., M.A.B., N.H., and A.Ma. performed all in vivo experiments. D.M.H. and J.N.B. performed surgical excision for acute slices. D.M.H., P.S.U., A.Me., N.H., A.Ma., K.R.C., T.S., and L.Y. prepared in vivo samples. F.Z., P.S.U., P.P., M.D.S., T.T.T.N., M.A.B., R.S., M.G.A., E.A. collected samples and performed proteomic, lipidomic and metabolomic analysis. P.S.U., D.M.H., B.R.S., P.C., J.N.B. interpreted results. P.S.U. and D.M.H. took the lead in writing the manuscript. All authors provided critical feedback to shape the research, analysis and manuscript.

## Competing interests

B.R.S., D.M.H., P.C., J.N.B., A.D., K.R.C., S.C., P.S.U. are inventors on patents and patent applications involving ferroptosis. B.R.S. co-founded and serves as a consultant to Inzen Therapeutics, Exarta Therapeutics, and ProJenX, Inc., serves as a consultant to Weatherwax Biotechnologies Corporation and Akin Gump Strauss Hauer & Feld LLP, and receives sponsored research support from Sumitomo Dainippon Pharma Oncology.
