## [Peer Review File · Nature Communications]

Dietary restriction of cysteine and methionine sensitizes gliomas to ferroptosis and induces alterations in energetic metabolismREVIEWER COMMENTS

Reviewer #1 (Remarks to the Author):

In this study, Upadhyayula, et al. provide data demonstrating that dietary restriction of cysteine and methionine (CMD) is a mechanism to sensitize gliomas to ferroptosis. They illustrate CMD synergizes with inhibition of GPX4 in in vitro and ex vivo organotypic models of GBM. The authors went on to show that CMD induces transcriptional hallmarks of ferroptosis, reduces oxidized glutathione levels and alters glioma cell metabolism in vitro, and modestly extends lifespan of glioma tumor-bearing mice. The in vitro metabolomic alterations in methionine-cysteine metabolism, taurine/hypo taurine metabolism and glutathione synthesis were shown to be translated in vivo. Lipidomic studies revealed a shift towards lipid profiles that are consistent with ferroptosis. Based on these results, the authors conclude that the CMD diet provides the context for therapeutic intervention by sensitizing gliomas, albeit modestly, to ferroptosis. Overall, this study is rigorously performed, and builds on a growing body of literature that attempts to harness the power of ferroptosis in oncology. The findings recapitulate published studies on ferroptosis; however, few new mechanistic insights are detailed as it relates to how ferroptosis is regulated and/or how this is impacted by dietary restriction. Indeed, improved longevity in mice upon restriction of sulfur-containing amino acids is a well appreciated phenomenon, including detailed study of the corresponding metabolic and energy alterations. The referee does appreciate that this has been studied to a lesser extent in cancer and is an important aspect of novelty in this study. Finally, the observed in vivo phenotype is modest and much less impactful than the ferroptosis sensitizing effect observed in vivo. This is not a criticism, per se, as differences between in vitro and in vivo ferroptotic sensitivity has emerged as an important question in the field. This had the potential to be an area for mechanistic novelty. For example, are other sulfur resources in vivo that can compensate for diet restriction?

Reviewer #2 (Remarks to the Author):

Review report:

In this manuscript authors exploit the unique role of cysteine in controlling the cell antioxidant response. To evaluate the impact of induced ferroptosis cell death in Glioblastoma cell lines and syngeneic orthotopic murine glioma model they combined Cyst/Met deprivation (CMD) and showed specific sensitization of RSL3-mediated cell death in both murine and human glioma cell lines and in ex-vivo slice cultures. Finally they report that Cyst/Met-restricted diet can improve survival in a syngeneic orthotopic murine glioma model.

This is a well executed and original study aimed to exploit possible actions of ferroptosis against cancer.

Critical points to address:

1- Authors demonstrate, as expected, that CMD specifically sensitize RSL3-mediated cell death (Fig1 A-F). However they should demonstrate that the partial loss of viability found in CMD cell lines is due to ferroptosis and prevented by Fer1.

Also it will be nice to compare and show the respective levels of mTORC1 and GPX-4 in CMD cells versus cells in normal medium.

2- The improved survival of syngeneic orthotopic murine glioma model in response to Cyst-Met-restricted diet (Fig 4B) is both interesting and intriguing.

Do the authors have indications that the mouse survival is a consequence of Glioma subjected to ferroptosis ? Any possible prevention by Fer1 in vivo to address this point ?

Reviewer #3 (Remarks to the Author):

Here, the authors demonstrate that (1) cysteine-methionine deprivation (CMD) sensitizes glioma

cells to ferroptosis inducers, (2) CMD improves survival in a mouse model of glioblastoma, and (3) CMD has profound impacts on glioma cells at the transcriptional, metabolic, and proteomic levels. The authors provide a very thorough characterization of the effects of CMD and make a good case for future work exploring the use of CMD in conjunction with ferroptosis-inducing drugs. This study is generally of high quality and claims are well supported.

As an aside, this is a compelling paper that would benefit from more attention being paid to the preparation and overall appearance of the figures (uniform fonts and font sizes, uniform line widths, etc.). While I am sure these will be fixed through the revision process, they are highly distracting (even on a first submission) from the otherwise high-quality nature of this manuscript.

In sum, I recommend this article to be accepted after certain concerns are addressed and missing pieces filled in.

Major comments

How do healthy cells respond to CMD? What is the IC50 of normal fibroblasts? Would transcription of ferroptosis-related genes and metabolic profiles also be altered under CMD diet in a non-glioma cell line? These data would be interesting in terms of overall safety of CMD (especially since in vivo experiments showed weight reduction in CMD mice) and in terms of quantifying how CMD sensitizes glioma cells (and whether different metabolic pathways are impacted in glioma vs healthy cells).

In line with this, Fig. 4F-G seem to show differences between healthy tissues under control and CMD. If full characterization of gene expression and metabolite levels (as in Fig. 2 and 3) cannot be done, could the authors at least use this IMS dataset to determine how CMD affects non-cancer cells? As is, it is not clear what the IMS data is adding to the manuscript, because there seems to be little exploration of spatial heterogeneity of metabolites in the tumors.

What was the overlap in metabolites identified by metabolite profiling and IMS? Did IMS reveal new any targets that were enriched in only portions of tumors (perhaps leading to modest overall changes that were not significant by metabolite profiling)?

What are the implications of this finding: "Interestingly, CMD led to a robust immunosuppressive signature involving downregulation of proteins related to antigen presentation and lymphocyte activation (Fig 4C)."? Could this be an issue for use of CMD if immunotherapy is being used in patient treatment?

Minor comments

First paragraph of the results, page 5 – can the authors please be more specific/explicit about the media formulation described in this sentence: "media was adapted for cell culture based on our previous ferroptosis permissive glioma culture methods."?

On page 5 - "In the primary ex vivo samples, CMD alone was sufficient to increase ROS levels." – why? Do the ex vivo samples include only cancer cells or surrounding non-cancer cells as well?

First paragraph of the results, page 5 – some of the figure callouts in the text seem to be missing – the dose response curves in Fig. 1B with Fer-1 are not referred to, for example.

Fig. 1D – it would be more helpful to include headings describing each panel than just including these details in the figure legend.

The logic behind this sentence: "Similar to the in vitro results, a low dose of RSL3 (100nM) plus CMD increased ROS to levels equivalent to a high dose of RSL3 (500nM)" should be introduced earlier on this page in the descriptions of Fig. 1D-E.

The heading "CMD leads to increased survival in vivo" on page 7 (and the results section beneath it) should more clearly and explicitly say that these data are for survival of a glioblastoma model.

Why do the lines extend further than the axes in Fig. 4B? (See comment below about overall figure quality)

On page 7 – should this comma be a period? "All tissue based analyses were performed on male mice to control for potential sex-specific metabolic effects, CMD induced alterations..."

Fig. 4H – the red and blue boxes require a legend. One can only deduce what they represent from the main text.

On page 9 – remove the apostrophe: "Many of the other compounds significantly decreased in CMD tumor's"

Reviewer #4 (Remarks to the Author):

Summary: Upadhyayula and colleagues present a study evaluating ferroptosis protection by cysteine/methionine metabolism in glioma. Their rationale is that cysteine that is consumed directly or produced via the methionine-dependent transsulfuration pathway is a requisite component of the glutathione tripeptide. Glutathione serves as a substrate for GPX4, an enzyme that blocks lipid peroxidation, thereby linking cysteine/methionine metabolism with ferroptosis resistance. This model is interesting and is supported by previous evidence in the literature tying cysteine deprivation to ferroptosis sensitization. The development of new treatment strategies for glioma is an urgent unmet clinical need, which this study directly addresses. The authors use appropriate in vitro and in vivo glioma models that are expected to reflect human glioma biology, including patient-derived organotypic slice cultures. My enthusiasm for this study is limited by a lack of appropriate controls in key experiments, insufficient experimental support for some conclusions drawn by the authors, and the absence of an in vivo study evaluating the combination of a cystine/methionine-reduced diet with a GPX4 inhibitor. This manuscript contains important findings and a compelling framework for future translational studies, but there are also several issues that weaken the authors' conclusions.

Major Points:

- 1) In Figures 1 and S1, the authors employ a panel of murine and human glioma cell lines that appear (although this is unclear from the limited information on culture conditions provided in the methods section) to be cultured in different media types. However, the authors use a single DMEM-based, cystine/methionine-depleted media for all cystine/methionine restriction experiments. Without additional information, it appears that multiple metabolites, including but not limited to cystine/methionine, are altered in the two culture conditions tested in Figures 1 and S1. This is particularly important because neurosphere cell lines, such as TS543, can be particularly sensitive to altered culture conditions. A potential impact of metabolites other than cystine and methionine on GPX4 inhibitor sensitivity cannot be ruled out based on the current experimental design.
- 2) In the Results section, the authors claim "Dose response assays demonstrated that RSL3 and ML-210 [...] both had synergistic enhancement of ferroptosis with CMD." A weakness of the data presented in Figures 1 and S1 is that the effect of cystine/methionine depletion on cell viability in the absence of other treatments is not shown. Without knowing the effect of each variable being tested, it's not possible to accurately assess synergy. The robust induction of oxidative stress by the CMD condition alone (Fig. 1F) leads me to believe that its impact on cell viability in the absence of GPX4 inhibition is not negligible.
- 3) The title of Figure 2 states "Cysteine methionine deprivation induces [...] an integrated stress response". Claims related to integrated stress response signaling pathway activation should be supported by eIF2 phosphorylation assays and/or by ISRIB treatment assays to verify the specificity of transcriptional changes shown in Figure 2 for this pathway.
- 4) The authors state, "The top upregulated metabolites (ascorbic acid, n-acetylputrescine, l-kynurenine, deoxyuridine) were closely tied to the citric acid cycle." These metabolites have indirect associations with the TCA cycle but the claim of a close tie is inaccurate. Moreover, the

TCA cycle pathway did not score highly in the pathway analysis in Figure 3B and effect sizes of changes in malate and fumarate (Fig. 3F) were modest in comparison to other metabolic changes. Taken together, these findings suggest that changes in oxidative metabolism may be secondary effects rather than primary consequences.

5) The authors claim that the "CMD diet is non-toxic, chronically tolerated [...]" yet show that mice on this diet rapidly (over ~20 days) lose weight relative to mice on normal chow. This finding is at odds with the conclusion and decreases enthusiasm for the translational relevance of this approach.

6) The authors set the stage by showing the potent ferroptotic effects of cystine/methionine restriction and GPX4 inhibition, but go on to test only the former in an in vivo model (Fig. 4B). The effect size is modest and the reader is left wondering whether the combined treatment a) is tolerable and b) elicits a more robust antitumor response than the dietary intervention alone.

7) The fact that cystine and methionine are not depleted in tumors in vivo by the CMD diet (Figure S2E) calls into question how physiological compensatory mechanisms may offset putative benefits of this diet. Also, lack of information on reduced glutathione content of tumors in mice on the CMD diet makes it difficult to discern how restriction of these amino acids affects substrate availability for GPX4.

Minor Points:

1) I believe the final 2 conditions for each cell line in Figure S1H are mislabeled.

2) If PTGS2 expression is an accurate biomarker of ferroptosis induction, why was it not evaluated in human models in Figure 2?

3) Showing H&E stains for parallel tissue sections in Fig. 4F would be helpful in evaluating tissue architecture and tumor margins.

RESPONSE TO REVIEWERS' COMMENTS

We thank the reviewers for their comprehensive review of our study. We have addressed all of the reviewer comments. To this end, we have performed several additional studies, including the following: in-vitro experiments examining the effects of RSL3 and CMD on healthy astrocytes, a confirmatory metabolomic analysis of in-vitro and in-vivo samples exposed to short term CMD (1 - 4 days), an in-vivo experiment combining CMD with local convection enhanced delivery of RSL3, and in-vivo expression of canonical ferroptosis markers after CMD and RSL3 co-treatment to demonstrate a tissue level response. We have also revised the text to better explain our findings and address specific points that were raised by the reviewers. Our hope is these new studies and revisions will answer the major reviewer critiques and also help further the understanding of the cysteine-methionine diet.

Below we provide a point-by-point response to all reviewers' comments. We have highlighted the corresponding revisions in the text of the manuscript.

Reviewer #1 (Remarks to the Author):

In this study, Upadhyayula, et al. provide data demonstrating that dietary restriction of cysteine and methionine (CMD) is a mechanism to sensitize gliomas to ferroptosis. They illustrate CMD synergizes with inhibition of GPX4 in in vitro and ex vivo organotypic models of GBM. The authors went on to show that CMD induces transcriptional hallmarks of ferroptosis, reduces oxidized glutathione levels and alters glioma cell metabolism in vitro, and modestly extends lifespan of glioma tumor-bearing mice. The in vitro metabolomic alterations in methionine-cysteine metabolism, taurine/hypotaurine metabolism and glutathione synthesis were shown to be translated in vivo. Lipidomic studies revealed a shift towards lipid profiles that are consistent with ferroptosis. Based on these results, the authors conclude that the CMD diet provides the context for therapeutic intervention by sensitizing gliomas, albeit modestly, to ferroptosis.

Overall, this study is rigorously performed, and builds on a growing body of literature that attempts to harness the power of ferroptosis in oncology. The findings recapitulate published studies on ferroptosis; however, few new mechanistic insights are detailed as it relates to how ferroptosis is regulated and/or how this is impacted by dietary restriction. Indeed, improved longevity in mice upon restriction of sulfur-containing amino acids is a well appreciated phenomenon, including detailed study of the corresponding metabolic and energy alterations. The referee does appreciate that this has been studied to a lesser extent in cancer and is an important aspect of novelty in this study. Finally, the observed in vivo phenotype is modest and much less impactful than the ferroptosis sensitizing effect observed in vivo. This is not a criticism, per se, as differences between in vitro and in vivo ferroptotic sensitivity has emerged as an important question in the field. This had the potential to be an area for mechanistic novelty. For example, are other sulfur resources in vivo that can compensate for diet restriction?

We thank the reviewer for their comments and time. To address the reviewers comments we have performed additional experiments including an animal experiment combining ferroptosis inducer RSL3 with the CMD diet, demonstrating a robust survival benefit (Fig 4C/D). We also agree with the importance of fine-tuning the in vivo diet to enhance the pro-ferroptotic effect, and think this is an exciting area of future research. Finally we added an experiment showing metabolomic profiling of tumors after 2 and 4 days of the CMD diet (compared to control). This data in Figure 4E further

highlights that even short term treatments can lead to tumor associated metabolomic changes highlighting the need for further studies for optimization of diet paradigms.

Reviewer #2 (Remarks to the Author):

Review report:

In this manuscript authors exploit the unique role of cysteine in controlling the cell antioxidant response. To evaluate the impact of induced ferroptosis cell death in Glioblastoma cell lines and syngeneic orthotopic murine glioma model they combined Cyst/Met deprivation (CMD) and showed specific sensitization of RSL3-mediated cell death in both murine and human glioma cell lines and in ex-vivo slice cultures. Finally they report that Cyst/Met-restricted diet can improve survival in a syngeneic orthotopic murine glioma model.

This is a well executed and original study aimed to exploit possible actions of ferroptosis against cancer.

We thank the reviewer for their time and consideration of our manuscript.

Critical points to address:

1- Authors demonstrate, as expected, that CMD specifically sensitizes RSL3-mediated cell death (Fig1 A-F). However they should demonstrate that the partial loss of viability found in CMD cell lines is due to ferroptosis and prevented by Fer1.

We think this comment addresses a critical point. It is important to note we do not believe that CMD induces ferroptosis selectively nor that the decrease in cell viability seen with the CMD media is due specifically to ferroptosis. Supplementary Figure 1E/F shows that even when CMD + ferrostatin-1 is added to values of 0 nM of drug (ML-210 or RSL3) it does not rescue cell viability to control levels. This shows that the effect of CMD is not purely through ferroptosis. However, our result also show that CMD sensitize glioma cells to ferroptosis, likely for multiple reasons including the reduction of the cysteine containing tripeptide glutathione.

To further demonstrate the synergistic effect of CMD and RSL3, we have also included synergy calculations between CMD and RSL3 for all cell lines across all doses of RSL3 studies in Supplementary Figure 1H.

Also it will be nice to compare and show the respective levels of mTORC1 and GPX-4 in CMD cells versus cells in normal medium.

We have attempted qPCR with GPX4 with multiple primers. All experiments have either failed QC or failed to show differences between CMD and control levels (across multiple cell lines MG1, MG2 and MG4). For these reasons we settled on GSH level/activity as a surrogate marker within CMD media versus control.

2- The improved survival of syngeneic orthotopic murine glioma model in response to Cyst-Met-restricted diet (Fig 4B) is both interesting and intriguing.

Do the authors have indications that the mouse survival is a consequence of Glioma subjected to ferroptosis ? Any possible prevention by Fer1 in vivo to address this point ?

We added another experiment involving the local delivery of RSL3 in-vivo to control versus CMD mice (Fig 4C/D). In this experiment the co-treatment had a durable increase in survival demonstrating the diet creates changes to the tumor that sensitize it to ferroptosis (by a specific ferroptosis/GPX4 inhibitor) in vivo.

Reviewer #3 (Remarks to the Author):

Here, the authors demonstrate that (1) cysteine-methionine deprivation (CMD) sensitizes glioma cells to ferroptosis inducers, (2) CMD improves survival in a mouse model of glioblastoma, and (3) CMD has profound impacts on glioma cells at the transcriptional, metabolic, and proteomic levels. The authors provide a very thorough characterization of the effects of CMD and make a good case for future work exploring the use of CMD in conjunction with ferroptosis-inducing drugs. This study is generally of high quality and claims are well supported.

As an aside, this is a compelling paper that would benefit from more attention being paid to the preparation and overall appearance of the figures (uniform fonts and font sizes, uniform line widths, etc.). While I am sure these will be fixed through the revision process, they are highly distracting (even on a first submission) from the otherwise high-quality nature of this manuscript.

We thank the reviewer for their time, interest and attention in reviewing our article. Their point is well taken; we have remade all figures in a unified file format to address any issues with formatting and fonts.

In sum, I recommend this article to be accepted after certain concerns are addressed and missing pieces filled in.

Major comments

How do healthy cells respond to CMD? What is the IC50 of normal fibroblasts?

We have addressed this concern by performing our dose response curve assays combining RSL3, Ferrostatin 1 and the CMD media in non-neoplastic astrocytes (Supplementary Figure 1J); astrocytes were chosen given their abundance within the brain and their function as support cells for the brain parenchyma. These cells were resistant to RSL3 and CMD showing minimal cell death even with combined treatment with both drug and amino acid deprivation.

Would transcription of ferroptosis-related genes and metabolic profiles also be altered under CMD diet in a non-glioma cell line? These data would be interesting in terms of overall safety of CMD (especially since in vivo experiments showed weight reduction in CMD mice) and in terms of quantifying how CMD sensitizes glioma cells (and whether different metabolic pathways are impacted in glioma vs healthy cells).

To address this, we performed additional analyses to assess the effects of CMD on non-neoplastic astrocytes (Supplementary Figure 1J) and normal brain tissue adjacent to gliomas (supplementary table 3), as noted above and below . The results suggest that normal tissues are less sensitive to CMD.

In line with this, Fig. 4F-G seem to show differences between healthy tissues under control and CMD. If full characterization of gene expression and metabolite levels (as in Fig. 2 and 3) cannot be done, could the authors at least use this IMS dataset to determine how CMD affects non-cancer cells?

We added supplementary table 3, which shows quantification of lipid profiles in non-tumor regions between CMD and control mice. Notably only 2 lipid species are significantly altered; none of them being the same as the ones altered within the tumor area. This shows that the CMD stress induces less significant changes within normal tissue; especially when combined with the astrocyte *in vivo* data.

As is, it is not clear what the IMS data is adding to the manuscript, because there seems to be little exploration of spatial heterogeneity of metabolites in the tumors.

We show in the IMS experiments that there is a spatial effect to CMD, suggesting different types of tissue in the brain have different responses to the CMD, particularly tumor and neighboring non-tumor tissue.

What was the overlap in metabolites identified by metabolite profiling and IMS? Did IMS reveal new any targets that were enriched in only portions of tumors (perhaps leading to modest overall changes that were not significant by metabolite profiling)?

IMS was used to visualize distribution of lipid species that were altered between CMD and control, while the bulk profiling studies examined metabolites and proteins. The overlap between metabolites and proteins was examined using the pathway analysis shown in Supplementary Figure 2C. We also added metabolite data from short-term CMD experiments (2 and 4 days) to show that alterations of glutathione, methionine and various sulfur containing metabolites are seen even in short treatments (Figure 4E) confirming that CMD has the intended *in vivo* effect.

What are the implications of this finding: “Interestingly, CMD led to a robust immunosuppressive signature involving downregulation of proteins related to antigen presentation and lymphocyte activation (Fig 4C).”? Could this be an issue for use of CMD if immunotherapy is being used in patient treatment? To address this comment, we have expanded our discussion on these finding, now in Supplementary figure 2G. “These findings suggest that CMD could affect patients’ response to immunotherapy”

Minor comments

First paragraph of the results, page 5 – can the authors please be more specific/explicit about the media formulation described in this sentence: “media was adapted for cell culture based on our previous ferroptosis permissive glioma culture methods.”?

We have expanded our methods section to be more explicit regarding our media formulation. To reflect this in the manuscript this sentence in the first paragraph of the results, page 5 now reads: “To this end, basal media was made either with normal DMEM or DMEM without cysteine and methionine. This process is fully described in the methods section and was adapted for cell culture based on our previous ferroptosis permissive glioma culture methods.^{13,15}”

On page 5 - "In the primary ex vivo samples, CMD alone was sufficient to increase ROS levels." – why? Do the ex vivo samples include only cancer cells or surrounding non-cancer cells as well? Given that slice cultures are taken from surgical specimens they include predominantly cancer cells, but also include non-neoplastic cells in the brain tumor microenvironment. The mechanism for CMD increasing ROS is likely similar to the effect that CMD alone has on cells in-vitro showing a decrease in glutathione levels.

First paragraph of the results, page 5 – some of the figure callouts in the text seem to be missing – the dose response curves in Fig. 1B with Fer-1 are not referred to, for example.

We have made substantial changes to the figures and the references in the text. All figures are now referenced in the text. Regarding this specific question the text now reads: "Ferrostatin, a ferroptosis inhibitor, prevented this lipid peroxidation. (Supp. Fig 1B) RSL3 mediated cell death, however, was not rescuable by necroptosis inhibitors (Nec-1s) or apoptosis inhibitors (ZVAD-FMK). (Supp. Fig 1C)"

Fig. 1D – it would be more helpful to include headings describing each panel than just including these details in the figure legend.

This has been done for Figure 1D and all similar figures throughout the revised manuscript .

The logic behind this sentence: "Similar to the in vitro results, a low dose of RSL3 (100nM) plus CMD increased ROS to levels equivalent to a high dose of RSL3 (500nM)" should be introduced earlier on this page in the descriptions of Fig. 1D-E.

To address this point, we added the following sentence to the description of Fig 1D,E. "Notably, a low dose (100nM) of RSL3 in combination with CMD increased lipid peroxidation to levels equivalent to a higher dose of RSL3 (500nM) (Fig. 1D, E). " We also added the following descriptor to Fig. 1F; "Similar to the *in vitro* results for lipid peroxidation, a low subthreshold dose of RSL3 (100nM) plus CMD increased ROS to levels equivalent to a high dose of RSL3 (500nM) (Control - 7.8%, 100nM RSL3 - 2.58%, 500nM RSL3 52.3%, CMD control - 29.7%, CMD + 100nM RSL3 - 53.5%; Fig 1F)."

The heading "CMD leads to increased survival in vivo" on page 7 (and the results section beneath it) should more clearly and explicitly say that these data are for survival of a glioblastoma model.

The first sentence of that results section has been modified to read " We next tested the effects of dietary CMD on survival in an orthotopic syngeneic murine glioma model."

Why do the lines extend further than the axes in Fig. 4B? (See comment below about overall figure quality)

We apologize for the quality of figures initially presented. All figures have been remade in adobe illustrator and this specific comment has been addressed to ensure the lines end at the axes for figure 4B.

On page 7 – should this comma be a period? "All tissue based analyses were performed on male mice to control for potential sex-specific metabolic effects, CMD induced alterations..."

Yes, this has been corrected.

Fig. 4H – the red and blue boxes require a legend. One can only deduce what they represent from the main text.

This has been added to the revised figure (panel 4I).

On page 9 – remove the apostrophe: “Many of the other compounds significantly decreased in CMD tumors”

This has been done.

Reviewer #4 (Remarks to the Author):

Summary: Upadhyayula and colleagues present a study evaluating ferroptosis protection by cysteine/methionine metabolism in glioma. Their rationale is that cysteine that is consumed directly or produced via the methionine-dependent transsulfuration pathway is a requisite component of the glutathione tripeptide. Glutathione serves as a substrate for GPX4, an enzyme that blocks lipid peroxidation, thereby linking cysteine/methionine metabolism with ferroptosis resistance. This model is interesting and is supported by previous evidence in the literature tying cysteine deprivation to ferroptosis sensitization. The development of new treatment strategies for glioma is an urgent unmet clinical need, which this study directly addresses. The authors use appropriate in vitro and in vivo glioma models that are expected to reflect human glioma biology, including patient-derived organotypic slice cultures. My enthusiasm for this study is limited by a lack of appropriate controls in key experiments, insufficient experimental support for some conclusions drawn by the authors, and the absence of an in vivo study evaluating the combination of a cystine/methionine-reduced diet with a GPX4 inhibitor. This manuscript contains important findings and a compelling framework for future translational studies, but there are also several issues that weaken the authors' conclusions.

We thank the reviewer for their comments. We agree that this study would be strengthened by an in vivo experiment testing the combination of a GPX-4 inhibitor and a CMD diet. We have performed this experiment and the co-treatment shows the expected result of a survival benefit with the co-treatment. This data, is presented in Figure 4D, shows that cotreatment with CMD diet and RSL3 local delivery leads to a significant and profound survival benefit over control diet with vehicle local delivery. We also address the additional concerns raised by this reviewer in a line-by line response below.

Major Points:

1) In Figures 1 and S1, the authors employ a panel of murine and human glioma cell lines that appear (although this is unclear from the limited information on culture conditions provided in the methods section) to be cultured in different media types. However, the authors use a single DMEM-based, cystine/methionine-depleted media for all cystine/methionine restriction experiments. Without additional information, it appears that multiple metabolites, including but not limited to cystine/methionine, are altered in the two culture conditions tested in Figures 1 and S1. This is particularly important because neurosphere cell lines, such as TS543, can be particularly sensitive to altered culture conditions. A potential impact of metabolites other than cystine and methionine on GPX4 inhibitor sensitivity cannot be ruled out based on the current experimental design.

For each set of experiments the only difference between the CMD media and control media was the concentration of cysteine and methionine. We have revised text to clarify this important point. To this end, we have added the following details about the media. “For all murine glioma cell lines cysteine methionine deprived media was made from basal DMEM without cysteine, methionine and glutamine (Thermo Fisher Scientific, 21013024) that was supplemented with L-glutamine to a final concentration of 4mM. All other components of the media were the same for control and CMD media. Human glioma

cells were cultured as previously described.^{37,38} TS543 cell neurosphere cell lines were cultured in Neurocult media as previously described,³⁷ but were dissociated and plated in a single cell monolayer in 96 well plates in either BFP or cysteine methionine deprived BFP for dose response experiments. KNS42 cell lines were cultured in DMEM+10%FBS or cysteine methionine deprived DMEM + 10%FBS. Thus, for all experiments the only difference between the CMD media and control media used for each cell line was the concentration of cysteine and methionine.”

For slice culture experiments the slices were cultured in a 1:1 mixture of DMEM and Hams-F12 with N-2 Supplement and 1% antimycotic/antibiotic. For treatment conditions, Hams-F12 without cysteine or methionine (MyBioSource, MBS652871) was mixed 1:1 with DMEM without cysteine or methionine (Thermo Fisher Scientific, 21013024) to make the DMEM/F12 without cysteine/methionine with the addition of N-2 supplement and 1% antimycotic/antibiotic.

Thus, all dose response/viability assays/slice culture experiments were performed with cysteine methionine being the singular variable altered between control and treatment groups.

2) In the Results section, the authors claim “Dose response assays demonstrated that RSL3 and ML-210 [...] both had synergistic enhancement of ferroptosis with CMD.” A weakness of the data presented in Figures 1 and S1 is that the effect of cystine/methionine depletion on cell viability in the absence of other treatments is not shown. Without knowing the effect of each variable being tested, it’s not possible to accurately assess synergy. The robust induction of oxidative stress by the CMD condition alone (Fig. 1F) leads me to believe that its impact on cell viability in the absence of GPX4 inhibition is not negligible. We agree with the reviewer concerns. The effect of CMD alone on cell viability is demonstrated by the 0nM RSL3 dose + CMD (blue curves in dose response curves in Fig. 1B). This shows that across the cell lines CMD alone at the 24 hour time point leads to a 20-40% reduction in cell viability alone. To accurately quantify synergy we used the coefficient of drug interaction formula which uses the effect size of one variable (CMD) compared to an effect size of a second variable (RSL3) and determines if the co-treatment is greater than the sum of two single treatments. By definition this uses the 0nM RSL3 +CMD as a comparison value. This quantification has been extended to all cell lines across RSL3 doses and is incorporated in Supp. Fig. 1H.

3) The title of Figure 2 states “Cysteine methionine deprivation induces [...] an integrated stress response”. Claims related to integrated stress response signaling pathway activation should be supported by eIF2 phosphorylation assays and/or by ISRIB treatment assays to verify the specificity of transcriptional changes shown in Figure 2 for this pathway.

We agree that the data does not support a change in the integrated stress response. We have removed this claim from the figure title and have focused on the finding related to markers of ferroptosis. The title for figure 2 now reads “ Cysteine methionine deprivation induces transcriptional hallmarks of ferroptosis”

4) The authors state, “The top upregulated metabolites (ascorbic acid, n-acetylputrescine, l-kynurenine, deoxyuridine) were closely tied to the citric acid cycle.” These metabolites have indirect associations with the TCA cycle but the claim of a close tie is inaccurate. Moreover, the TCA cycle pathway did not score highly in the pathway analysis in Figure 3B and effect sizes of changes in malate and fumarate (Fig. 3F) were modest in comparison to other metabolic changes. Taken together, these findings suggest that changes in oxidative metabolism may be secondary effects rather than primary consequences.

We have softened this statement to simply state what was observed. The revised text now states “Effects of CMD at 24 and 48 hours deprivation was examined (Fig 3B-D). Significant reductions in concentrations of hypotaurine, methionine, glutamine and glutamate were noted at 24 hours (Fig. 3B); at 48 hours both oxidized and reduced glutathione, glutamate, and hypotaurine/taurine were also decreased (Fig. 3D). Significant increases in several amino acids and metabolites were seen at both 24 and 48 hours (Fig. 3B, E).”

5) The authors claim that the “CMD diet is non-toxic, chronically tolerated [...]” yet show that mice on this diet rapidly (over ~20 days) lose weight relative to mice on normal chow. This finding is at odds with the conclusion and decreases enthusiasm for the translational relevance of this approach.

This comment is addressed by the new data for the survival studies in Figure 4. Notably, several of the mice on the CMD diet survived for longer than 100 days while on the CMD diet, highlighting that the diet can be tolerated for several months.

6) The authors set the stage by showing the potent ferroptotic effects of cystine/methionine restriction and GPX4 inhibition, but go on to test only the former in an *in vivo* model (Fig. 4B). The effect size is modest and the reader is left wondering whether the combined treatment a) is tolerable and b) elicits a more robust antitumor response than the dietary intervention alone.

As described above we have addressed this comment by adding a new survival study with combination treatment showing a robust survival benefit with co-treatment of local delivery of RSL3 and CMD diet (median survival 112 days) compared to local delivery of vehicle and a control diet (median survival 56 days), $p=0.01$.

7) The fact that cystine and methionine are not depleted in tumors *in vivo* by the CMD diet (Fig. S2E) calls into question how physiological compensatory mechanisms may offset putative benefits of this diet. Also, lack of information on reduced glutathione content of tumors in mice on the CMD diet makes it difficult to discern how restriction of these amino acids affects substrate availability for GPX4.

We addressed this issue with additional experimentation. Given that *in vitro* data occurred in the 24-48 hour CMD timeframe, we aimed to setup a short term CMD *in vivo* experiment to examine tumor related metabolite changes in an acute deprivation (Fig. 4E). We found that at 2 and 4 days CMD *in vivo* led to alterations of glutathione, acetylmethionine, adenosyl-homocysteine and methionine. Notably, some of these differences lost significance between 2 and 4 days, pointing to possible compensatory mechanisms. These compensatory mechanisms should be the topic of future studies to determine the optimal timing of synergistic treatment.

Minor Points:

1) I believe the final 2 conditions for each cell line in Figure S1H are mislabeled.

A new figure S1 has been created to rectify issues with labeling.

2) If PTGS2 expression is an accurate biomarker of ferroptosis induction, why was it not evaluated in human models in Figure 2?

PTGS2 was evaluated in the human slice culture models but showed no significant changes from control. Notably, previous studies including Stockwell et al. 2017(28985560) show different temporal

relationships between CHAC1 and PTGS2 upregulation in response to ferroptotic stress. It is possible that our time point is inopportune for identification of both transcripts simultaneously.

3) Showing H&E stains for parallel tissue sections in Fig. 4F would be helpful in evaluating tissue architecture and tumor margins.

Unfortunately, no tissue is available for showing H&E stains in parallel section from the mice used for the DESI-IMS analyses (now shown in Fig. 4G,H).

REVIEWER COMMENTS

Reviewer #3 (Remarks to the Author):

The Authors have addressed all of the concerns raised in the original review.

Reviewer #4 (Remarks to the Author):

The authors have addressed some of my comments in this revision. Notably, the combination therapy study involving CMD and RSL3 treatment modalities strengthens the paper considerably. However, some points were not addressed through this revision and certain passages do not accurately reflect the underlying data in the manuscript. Please see the points below for descriptions of these issues.

Major Points:

1) The authors still fail to provide information on the toxicity of CMD alone in the cell lines used in Figures 1 and S1. For example, the authors state in their rebuttal that "The effect of CMD alone on cell viability is demonstrated by the 0nM RSL3 dose + CMD (blue curves in dose response curves in Fig. 1B)." However, the blue curve referenced is RSL3+CMD and the x-axis is in log scale, meaning that the 0 nM RSL3 dose (i.e. CMD alone) is not displayed. Readers cannot evaluate synergy without knowing the effect of each treatment alone.

2) I have concerns regarding the validity of the synergy calculation now included as Fig. S1H. To cite one example, the 250 nM dose of RSL3 is shown to be highly synergistic with CMD in MG3 cells (data from Fig. 1B). However, 250 nM RSL3 alone reduces viability by nearly 100% in this line (Fig. 1B). If this synergy calculation method is unreliable, the authors should show all the data for each cell line represented in Fig. S1H, including the effect of each treatment alone. This is an important issue because claims of synergy between CMD and ferroptosis induction are a centerpiece of this study.

3) The authors write in the manuscript "The diet was tolerated with no adverse effects, though notably CMD mice maintained lower weights than control mice (Supp. Fig 2D)." The authors do not show data pertaining to any toxicity biomarkers they evaluated other than body weight. The fact remains that body weights were significantly decreased on the CMD diet and body weight is an established marker of toxicity in mice. Many treatments prolong survival in mice but have toxicity profiles that would not support testing in patients. The bar cannot be set so low for preclinical therapeutic studies in cancer research that the absence of death is equated with "no adverse effects", as the authors allude to in their rebuttal (point #5). The authors should revise the sentence in the manuscript to acknowledge that there may be issues with toxicity of the CMD diet, as this represents a key issue to address in future studies that may build on the translational work in this manuscript.

4) The absence of histology analysis of the tissues evaluated by DESI-IMS in Fig. 4G and 4H is problematic and was not addressed. The authors observe widespread regional variance in phospholipid abundance independent of treatment. How can these data be interpreted if the readers do not know whether the regional variance is associated with non-malignant tissue present in the specimen, necrotic areas of the tumor, or other factors related to tissue architecture? At a minimum, the authors should acknowledge this limitation in the manuscript by commenting on the difficulties that the lack of histology present for data interpretation in these panels.

Reviewer #5 (Remarks to the Author):

I have been asked to comment on the revised manuscript. This is a novel study that introduces a new concept for sensitizing to ferroptosis in vivo. The authors appear to have addressed all reviewing comments with new data, analyses and clarifications. This is work that should be

published swiftly even if all mechanistic details of the effect are unclear.

Two minor points on the figures:

Figure 1D: substitute the letter 'u' in μM for the character micron (μ).

Figure 2 and elsewhere: it would be more useful to report individual datapoints from the separate experiments than mean \pm SEM

RESPONSE TO REVIEWERS' COMMENTS

Reviewer #3 (Remarks to the Author):

The Authors have addressed all of the concerns raised in the original review.

We thank the reviewer for their time and energy in reviewing the manuscript.

Reviewer #4 (Remarks to the Author):

The authors have addressed some of my comments in this revision. Notably, the combination therapy study involving CMD and RSL3 treatment modalities strengthens the paper considerably. However, some points were not addressed through this revision and certain passages do not accurately reflect the underlying data in the manuscript. Please see the points below for descriptions of these issues.

We thank the reviewer for their time and comments that have improved the manuscript.

Major Points:

1) The authors still fail to provide information on the toxicity of CMD alone in the cell lines used in Figures 1 and S1. For example, the authors state in their rebuttal that “The effect of CMD alone on cell viability is demonstrated by the 0nM RSL3 dose + CMD (blue curves in dose response curves in Fig. 1B).” However, the blue curve referenced is RSL3+CMD and the x-axis is in log scale, meaning that the 0 nM RSL3 dose (i.e. CMD alone) is not displayed. Readers cannot evaluate synergy without knowing the effect of each treatment alone.

We appreciate this important point. To clarify, a dose response curve with the 0nM RSL3 dose was displayed in Supplementary Figure 1E in our previous submission. To further address this point, we have revised Supplementary Figure 1 to now include the dose response curves with the 0nM RSL3 across all the cell lines used for synergy calculations (new Supplementary Figure 1 E-J).

2) I have concerns regarding the validity of the synergy calculation now included as Fig. S1H. To cite one example, the 250 nM dose of RSL3 is shown to be highly synergistic with CMD in MG3 cells (data from Fig. 1B). However, 250 nM RSL3 alone reduces viability by nearly 100% in this line (Fig. 1B). If this synergy calculation method is unreliable, the authors should show all the data for each cell line represented in Fig. S1H, including the effect of each treatment alone. This is an important issue because claims of synergy between CMD and ferroptosis induction are a centerpiece of this study.

We thank the reviewer for highlighting this concern. To clarify, the synergy calculations were conducted on dose response curves calculated from three independent replicate experiments each with a triplicate of each treatment condition/dose. This data is now shown in Supplementary Figure E-J. These results show

that CMD enhances RSL3-induced cell death across all cell lines. Additionally, we performed the synergy calculations using the established Chou-Talaly method at each dose of RSL3. These calculations show that CMD is synergistic with RSL3 across a range of doses, including the lowest dose of RSL3 tested (62.5 nM) (new Supplementary Figure 1M).

3) The authors write in the manuscript “The diet was tolerated with no adverse effects, though notably CMD mice maintained lower weights than control mice (Supp. Fig 2D).” The authors do not show data pertaining to any toxicity biomarkers they evaluated other than body weight. The fact remains that body weights were significantly decreased on the CMD diet and body weight is an established marker of toxicity in mice. Many treatments prolong survival in mice but have toxicity profiles that would not support testing in patients. The bar cannot be set so low for preclinical therapeutic studies in cancer research that the absence of death is equated with “no adverse effects”, as the authors allude to in their rebuttal (point #5). The authors should revise the sentence in the manuscript to acknowledge that there may be issues with toxicity of the CMD diet, as this represents a key issue to address in future studies that may build on the translational work in this manuscript.

We have revised the manuscript to acknowledge that there are potential issues with toxicity of the CMD diet. To this end we have revised our claims related to this point in two places in the paper. In the results section we now write:

“The diet was tolerated throughout the duration of the survival study, although notably mice on CMD diet maintained lower weights than control mice (Supp. Fig 2D)”.

In the discussion we now write:

“Here we show that CMD diet was tolerated throughout the duration of the survival studies and was associated with a significant survival benefit, indicating local effects on brain tumor growth and response to RSL3. Notably, the mice on the CMD diet maintained a significantly lower weight, and there may be issues with systemic toxicity that will need to be addressed in future studies.”

4) The absence of histology analysis of the tissues evaluated by DESI-IMS in Fig. 4G and 4H is problematic and was not addressed. The authors observe widespread regional variance in phospholipid abundance independent of treatment. How can these data be interpreted if the readers do not know whether the regional variance is associated with non-malignant tissue present in the specimen, necrotic areas of the tumor, or other factors related to tissue architecture? At a minimum, the authors should acknowledge this limitation in the manuscript by commenting on the difficulties that the lack of histology present for data interpretation in these panels.

To acknowledge this limitation, we have added the following to our discussion:

“Notably, the tumor microenvironment is heterogenous, and regional variance associated with necrosis, microvascular proliferation and infiltrated brain tissue may contribute to the patterns of phospholipid abundance. Future studies are needed to determine the effects of CMD on the brain tumor microenvironment and to assess whether they are therapeutically actionable.”

Reviewer #5 (Remarks to the Author):

I have been asked to comment on the revised manuscript. This is a novel study that introduces a new concept for sensitizing to ferroptosis in vivo. The authors appear to have addressed all reviewing comments with new data, analyses and clarifications. This is work that should be published swiftly even if all mechanistic details of the effect are unclear.

We thank the reviewer for their time and consideration.

Two minor points on the figures:

Figure 1D: substitute the letter 'u' in uM for the character micron (μ).

We have remade Figure 1D to address this comment.

Figure 2 and elsewhere: it would be more useful to report individual datapoints from the separate experiments than mean +/- SEM

We have remade Figure 2 to address this comment.

REVIEWERS' COMMENTS

Reviewer #4 (Remarks to the Author):

The authors have addressed all of my concerns.